# BLACK-BOX PRIVACY ATTACKS ON SHARED REPRESENTATIONS IN MULTITASK LEARNING

**John Abascal**[*]**, Alina Oprea & Jonathan Ullman**
Northeastern University
{abascal.j, a.oprea, j.ullman}@northeastern.edu

**Nicolás Berrios**[†]
Georgia Institute of Technology
berrios@gatech.edu

**Adam Smith**
Boston University & Google Deepmind
ads22@bu.edu

**Matthew Jagielski**
Google Deepmind
jagielski@anthropic.com

[*]Corresponding author.    [†]Work completed while visiting Northeastern University.

## ABSTRACT

The proliferation of diverse data across users and organizations has driven the development of machine learning methods that enable multiple entities to jointly train models while minimizing data sharing. Among these, *multitask learning* (MTL) is a powerful paradigm that leverages similarities among multiple tasks, each with insufficient samples to train a standalone model, to solve them simultaneously. MTL accomplishes this by learning a *shared representation* that captures common structure between tasks and generalizes well across them all. Despite being designed to be the smallest unit of shared information necessary to effectively learn patterns across multiple tasks, these shared representations can inadvertently leak sensitive information about the particular tasks they were trained on.

In this work, we investigate privacy leakage in shared representations through the lens of inference attacks. Towards this, we propose a novel, *black-box task-inference* threat model where the adversary, given the embedding vectors produced by querying the shared representation on samples from a particular task, aims to determine whether the task was present in the multitask training dataset. Motivated by analysis of tracing attacks on mean estimation over mixtures of Gaussian distributions, we develop efficient, purely black-box attacks on machine learning models that exploit the dependencies between embeddings from the same task without requiring shadow models or labeled reference data. We evaluate our attacks across vision and language domains when MTL is used for personalization and for solving multiple distinct learning problems, and demonstrate that even with access only to fresh task samples rather than training data, a black-box adversary can successfully infer a task's inclusion in training. [1]

## 1 INTRODUCTION

*Multitask learning* (MTL) has emerged as a powerful paradigm that leverages similarities among multiple learning tasks, each with insufficient samples to train a standalone model, to solve them simultaneously while minimizing data sharing across multiple entities, such as users and organizations. MTL accomplishes this goal by learning a *shared representation* that captures common structure between the tasks. Concretely, a shared representation could be a neural network that learns a mapping of the data from all tasks into a shared feature space, where similar data points across tasks cluster together, and task-specific output layers (or heads) operate on these embeddings—that is, the outputs

---

[1]Code available at https://github.com/johnmath/task-inference-attacks

of the shared representation—to make predictions. MTL methods have shown remarkable success in various domains, including computer vision (Girshick, 2015), natural language processing (Collobert & Weston, 2008), federated learning (Smith et al., 2017; Hanzely & Richtarik, 2021; Mansour et al., 2020; Ghosh et al., 2020), drug discovery (Ramsundar et al., 2015), and financial forecasting (Ghosn & Bengio, 1996). For example, MTL can be used to personalize image classification models by learning a shared representation across many users and locally adapting to each user's small, on-device photo library.

Despite being designed to capture only generic patterns that can be applied to several downstream tasks, these shared representations can inadvertently leak sensitive information about specific tasks, or underlying data distributions, that they were trained on. The privacy risks that arise are of particular concern when data from several sensitive entities, such as individual users, private organizations, or data silos, are jointly used for training across multiple tasks or non-uniform data distributions. Here, shared representations are often the minimum piece of information that each entity must contribute in order to achieve strong generalization while only contributing a limited number of samples. This point is highlighted in prior work on privacy-preserving collaborative learning (Shen et al., 2023), where sharing *only* the representation, not task-specific layers, demonstrably improves convergence rates and model performance when learning over multiple parties.

In our work, we study privacy attacks at the level of entire tasks and broadly investigate the privacy risks of jointly learning over multiple tasks by mounting these attacks on the smallest unit of information required to jointly train a model, the shared representation. In particular, we attempt to determine whether, and to what extent, an adversary can infer a given task's inclusion in training given black-box query access to (i.e. the ability to sample embedding vectors from) the shared representation. While there is prior work that explores privacy attacks on MTL, this work is limited to sample-level membership inference and model extraction (Yan et al., 2024) where the adversary can make queries to the task-specific heads of the multitask model and train reference models.

Towards this goal, we propose a new threat model called *task-inference* and develop black-box attacks with minimal adversarial knowledge to infer the inclusion of a task in MTL. Unlike membership-inference (Homer et al., 2008; Sankararaman et al., 2009; Dwork et al., 2015; Shokri et al., 2016), task-inference generalizes individual privacy attacks from the sample level to the task level, where the adversary aims to determine the presence of an entire target task, rather than a particular sample, in the training set. Critically, our threat model only assumes that the adversary has access to samples from the target task's distribution. We identify two variants of our threat model, one with a **strong** adversary who has the specific samples used to train the shared representation and another with a **weak** adversary who only has independent samples from the target task's distribution. Our experiments demonstrate a notable separation between these two threat models in terms of their attack capabilities, and we provide analysis of a simplified learning setting to help explain our empirical findings in machine learning. We observe that while both adversaries can mount successful attacks, having access to training samples provides a sizeable advantage. Furthermore, in contrast to prior work (Shokri et al., 2016; Carlini et al., 2022; Liu et al., 2021a; Chaudhari et al., 2022; Abascal et al., 2023; Hartmann et al., 2023; Yan et al., 2024), neither of the task-inference adversaries require auxiliary datasets to train reference, or shadow, models to calibrate their attack. Instead, our attack exploits the key observation that embeddings from the same task are codependent, which allows an adversary with multiple samples to amplify the membership signal and construct a powerful statistical test without reference models.

We comprehensively evaluate our attacks across vision (CelebA (Liu et al., 2015)) and language (Stack Overflow (Annamoradnejad et al., 2022)) datasets for two representative MTL use cases. The first is *personalization*, where each user is a task with few samples for a single, shared learning objective (e.g. a personalized spam detector). The second is solving *multiple learning problems* simultaneously, where each task is a distinct classification problem with insufficient data to train a standalone model, but whose shared underlying structure allows them to be solved jointly (e.g., detecting different post topics). The privacy risks of task-inference are therefore contextual: in personalization, the attack leaks an individual's participation in training, while for multiple learning problems, it reveals the inclusion of an entire subpopulation. We also study the factors that lead to task-inference leakage by observing how attack success varies in the model's generalization gaps. Our findings demonstrate that a purely black-box adversary can successfully infer inclusion of tasks in MTL using only query access to the shared representation, even with **weak** access to training data samples.

## 2 BACKGROUND AND RELATED WORK

**Multitask Learning.** The goal of MTL (see (Caruana, 1997) for an early survey) is to learn jointly over several related tasks, often with sparse data, by exploiting shared features between them. A *task* $\tau$ is defined as a distribution over samples in $\mathcal{X} \times \mathcal{Y}$ (e.g. $d$-dimensional real vectors that correspond to binary labels). In this work, we assume that the tasks for some learning problem are drawn from a *task distribution*, $\mathcal{Q}$. To leverage common features between tasks, we can learn a shared representation $h : \mathcal{X} \to \mathcal{Z}$, that maps samples in $\mathcal{X}$ to a lower dimensional space, $\mathcal{Z}$, where the representation vectors, called *embeddings*, capture the shared structure of the tasks. Mapping these samples to a lower dimensional space simplifies the learning problem, allowing us to use simple, linear classifiers $g_i : \mathcal{Z} \to \{0, 1\}$. Given data samples from $T$ tasks $\vec{\tau} = (\tau_1, \ldots, \tau_T)$ and a loss function $\mathcal{L}$, learning the shared representation $h$ can be written as the following optimization problem:

$$\min_{h_\theta \in \mathcal{H}} \left( \min_{g_{\beta_1}, \ldots, g_{\beta_T}} \sum_{i=1}^{\mathcal{T}} \mathcal{L}(g_{\beta_i} \circ h_\theta, \, \tau_i) \right)$$

The shared representation $h$ often takes the form of a neural network that maps an example $x \in \mathcal{X}$ to an embedding vector $z \in \mathcal{Z}$ with task-specific classifiers $\{g_1, ..., g_T\}$ being linear classifiers applied to the embedding vector. Thus, the goal is to learn an $h$ such that it maps data samples to embedding vectors that are linearly separable.

In this work, we consider two MTL settings that are representative of common scenarios: (1) *personalization*, where each *user* or *person* represents a task with a personalized objective (e.g., multilabel face detection, recommendation, etc.) learned from sparse data by leveraging the shared representation trained over all users; and (2) *multiple learning problems*, where the tasks correspond to distinct classification problems (e.g., detection of facial attributes, binary topic classification, etc.) that share a similar underlying structure. We choose these two settings because they highlight how our proposed threat model (Section 3) captures and abstracts the privacy threats posed by existing attacks at the individual level (Shokri et al., 2016), user level (Kandpal et al., 2023), and property level (Ateniese et al., 2015).

MTL has deep connections to federated learning (FL) (McMahan et al., 2017), a machine learning framework where a central, trusted server coordinates model training across several parties, such as users or silos (Huang et al., 2022). Several works study the connections between FL and MTL (Smith et al., 2017; Yu et al., 2022; Tan et al., 2023; Hu et al., 2023; Fallah et al., 2020; Mansour et al., 2020), many of which focus on techniques to learn personalized models by locally adapting a shared representation on-device. The widespread use of FL (Lu & Wang, 2022) serves as additional motivation for studying privacy leakage from the shared components of collaborative learning.

**Privacy Attacks on ML Models.** While overparameterized models are known to memorize training data (Zhang et al., 2017), they often do so without compromising generalization (Belkin et al., 2019; Feldman, 2020), making privacy attacks a significant threat (Shokri et al., 2016). These risks are typically studied through membership-inference attacks (or MIAs), which represent a fundamental form of privacy leakage. The goal of an MIA adversary is to determine whether specific record was used in training (Homer et al., 2008). We generalize this notion of leakage by defining a novel threat model for MTL that restricts the adversary's access to the shared representation, as it constitutes the minimal information that must be exchanged for jointly training over tasks with few samples. One work (Yan et al., 2024) adapts sample-level MIAs to the MTL setting but assumes query access to the task-specific classification heads; this assumption would render our goal of task-level inference trivial.

Our proposed threat model is distinct in that it generalizes and interpolates several existing privacy attacks that operate at a coarser granularity than the sample-level depending on the definition of tasks within MTL. Namely, property (Ateniese et al., 2015; Chaudhari et al., 2022; Hartmann et al., 2023) and dataset (Maini et al., 2024) inference aim to determine the frequency and inclusion of entire subpopulations in the training data of a machine learning model, respectively. Similarly, the goal of a user-inference (Kandpal et al., 2023) adversary is to infer whether a user's entire data contribution was present in training. In contrast to these works, we target the intermediate embedding directly, rather than the final model, and assume minimal access to side information like reference data.

Multiple works have also studied attacks on representations under different assumptions. Existing attacks on representations (Song & Raghunathan, 2020; Liu et al., 2021a) target models that are trained with contrastive loss, which explicitly maximizes the separation between embeddings of different samples. Our work differs from these approaches by investigating leakage of sensitive information from representations which are learned implicitly to generalize across a diverse set of supervised, downstream tasks.

## 3 THREAT MODEL

We investigate whether, and to what degree, shared representations leak information about the specific tasks they were trained on through the lens of inference attacks (Homer et al., 2008). In this setting, there is an MTL model that is trained on several tasks simultaneously to learn a shared representation, or encoder, and individual task layers. Black-box queries can then be made to the encoder to receive representation vectors, or embeddings, for any given input. A challenge task is drawn from the same distribution as the tasks used to train the MTL model. Our adversary uses their query access, along with data drawn from the challenge task, to infer whether the challenge task was included while training the MTL model. In this work, we study this leakage from a purely black-box perspective, assuming no knowledge of the underlying data distributions to train shadow models, as is common in prior works (Carlini et al., 2022). We can describe our threat model using the following security game between a *challenger* and an *adversary*:

**Task-Inference Security Game**

1. The challenger receives $T$ tasks $\vec{\tau} = (\tau_1, \ldots, \tau_T)$, drawn from a task distribution $\mathcal{Q}$. For each of the tasks, $\tau_i$, the challenger is given a batch of samples $X_i$ and concatenates the $T$ batches into a dataset $D = \{X_{\tau_1}, \ldots, X_{\tau_T}\}$.
2. The challenger trains a *shared representation* $h_\theta \leftarrow \mathcal{T}_{MTL}(D)$ by simultaneously learning *task-specific* models that share $h$.
3. The challenger randomly selects $b \in \{0, 1\}$. If $b = 0$, the challenger samples a challenge task $\tau^*$ from $\mathcal{Q}$ uniformly at random, such that $\tau^* \notin \vec{\tau}$. Otherwise, the challenger samples $\tau^*$ from $\vec{\tau}$ uniformly at random.
4. The challenger sends a batch of samples, $X^*$, drawn from the challenge task, $\tau^*$.
5. The adversary, using the batch of samples and black-box access to $h_\theta$, guesses a bit $\hat{b} \leftarrow \mathcal{A}(h_\theta(X_{\tau^*}))$.
6. The adversary wins if $\hat{b} = b$ and loses otherwise.

A key aspect of this security game is that the adversary only requires samples drawn from the challenge task, rather than the specific training samples from $D$. We claim that, even with *fresh, unseen data* from a challenge task, a task-inference adversary can successfully infer a task's inclusion in multitask training. We use the following terms to define this distinction: A task-inference adversary is **strong** if they have access to training samples when $b = 1$ or **weak** if they *do not* have access to training samples when $b = 1$.

Our proposed threat model is general and interpolates several existing attack types depending on the contextual definition of tasks. When tasks correspond to unique users in the dataset, task-inference leaks the same information as a user-inference (Kandpal et al., 2023) attack. If we further specify that each user has at most one training example, our threat model reduces to membership-inference. When tasks are defined by labeling or learning problem, task-inference mirrors property or dataset-inference (Chaudhari et al., 2022; Hartmann et al., 2023; Maini et al., 2024).

We note that our instantiation of the task-inference adversary operates in a purely black-box manner. In contrast to most existing work on membership and property-inference (Shokri et al., 2016; Carlini et al., 2022; Mahloujifar et al., 2022; Liu et al., 2021b; Abascal et al., 2023; Chen et al., 2023), our adversary does not require metaclassifiers or shadow models to mount their attack. The power of our approach comes from receiving a *set of correlated samples* from the challenge task, rather than just a single sample. By composing a test statistic over multiple embeddings, the membership signal is amplified, allowing the adversary to distinguish between **IN** and **OUT** tasks without additional information.

## 4 ANALYZING TASK-INFERENCE

To motivate further exploration of attacks on deep learning models, we present a simplified analog of multitask learning via mean estimation. A more detailed treatment of the problem and proofs of our theorems in this section can be found in Appendix B.2.

Let $M$ be the set of *task means* $\{\mu_1, \ldots, \mu_T\}$ which are sampled i.i.d. from $\mathcal{N}(\bar{\mu}, \bar{\sigma}^2 \mathbb{I}_d)$ where $\bar{\mu}$ is the "true" mean that we would like to estimate. For each "task" $\mu_i$, we can sample $N < T$ vectors i.i.d. from $\mathcal{N}(\mu_i, \sigma^2 \mathbb{I}_d)$ and store them in the set $X_i = \{x_{i,1}, \ldots, x_{i,N}\}$. Since no individual dataset $X_i$ would yield an accurate estimate of $\bar{\mu}$ we can instead compute the sample "multitask" mean using all of the $X_i$'s as $\hat{\mu} = \frac{1}{T} \sum_{i=1}^{T} \left( \frac{1}{N} \sum_{j=1}^{N} X_{i,j} \right)$. We note that $\hat{\mu}$ is Gaussian with expectation $\mathbb{E}_{\mu,X} [\hat{\mu}] = \bar{\mu}$ and covariance $\mathrm{Cov}(\hat{\mu}) = \left( \frac{\bar{\sigma}^2}{T} + \frac{\sigma^2}{N \cdot T} \right) \cdot \mathbb{I}_d$

We construct an adversary, based on prior work (Dwork et al., 2015), with similar variants to the **strong** and **weak** adversaries detailed in Section 3. We consider a challenger $\mathcal{C}$ who releases the statistic $\hat{\mu}$ and an adversary $\mathcal{A}$ who wants to learn whether a given task $\mu_i$ was included in the dataset that was used to compute $\hat{\mu}$.

1. $\mathcal{C}$ flips a coin $b \in \{0, 1\}$ and sends $\mathcal{A}$ a challenge set of $k < N$ samples from a training task $\mu_{IN} \in M$ (if $b = 1$) or a new, unseen task $\mu_{OUT} \sim \mathcal{N}(\bar{\mu}, \bar{\sigma}^2 \mathbb{I}_d)$ (if $b = 0$)
2. $\mathcal{A}$ takes the sample mean of their samples $\mu_B = \frac{1}{k} \sum_{x \in B} x$
3. $\mathcal{A}$ computes the test statistic $z = \langle \hat{\mu} - \bar{\mu} , \ \mu_B - \bar{\mu} \rangle$
4. $\mathcal{A}$ picks a threshold $\gamma$ and returns $\hat{b} = \mathbb{1}_{z > \gamma}$

This attack adapts a test statistic originally intended for membership-inference which measures the correlation between the released statistic and the adversary's samples. Unlike the membership-inference setting, the adversary is attempting to detect traces of the *task*, or data's distribution, $\mu_i$, rather than any particular sample itself. In the following theorems, we find that the inclusion of a task, $\mu_i$, can be inferred by both the **strong** and **weak** adversaries, and the **strong** adversary gets a slight advantage from having access to samples in $X_i$.

**Theorem 4.1** (Strong Adversary). *Let $\tau$ be the index of the target task and suppose that the challenger sends the adversary a set of $k$ samples such that, when the task is **IN**, the $k$ samples are drawn uniformly at random from $X_\tau$. Then, when $\mu_\tau$ is **OUT** and **IN**, respectively, we have*

$$\mathbb{E}_{\mu, X} [z_{OUT}] = 0 \quad and \quad \mathbb{E}_{\mu,X} [z_{IN}] = \frac{d}{T}(\bar{\sigma}^2 + \frac{\sigma^2}{N}).$$

Theorem 4.1 shows that the **strong** adversary's test statistic, $z$, grows in the dimension of the data divided by the total number of tasks used to learn the multitask mean, $\hat{\mu}$. This result demonstrates that detecting traces of tasks can be seen as a generalization of membership inference, as setting the number of tasks $T = 1$, sampling $N$ datapoints, $X$, only from the single task to estimate the task mean, and allowing the adversary to use $k = 1$ sample reduces $z_{IN}$ to the membership-inference test statistic, which has expectation $\Theta(\frac{d}{N})$. In our setting, the test statistic $z$ of an adversary who wants to trace a single example would be $\Theta(\frac{d}{NT})$.

**Theorem 4.2** (Weak Adversary). *Let $\tau$ be the index of the target task and suppose the challenger sends the adversary a set of $k$ samples such that, when the task is **IN**, the $k$ samples are drawn i.i.d. from the same distribution as $X_\tau$, $\mathcal{N}(\mu_\tau, \sigma^2 \mathbb{I}_d)$. Then the expected value of the statistic, $z$, when $\mu_\tau$ is **OUT** is 0. When $\mu_\tau$ is **IN**: $\mathbb{E}_{\mu,X} [z_{IN}] = \frac{d}{T} \bar{\sigma}^2$*

Informally, Theorem 4.2 shows that the expectation of the weak adversary's test statistic depends only on the number of tasks, and does not benefit from a smaller number of total training samples.

**Theorem 4.3** (Variance of $z$ (Informal)). *The variance of $z$ when $\mu_\tau$ is **IN** or **OUT** is*

$$\mathrm{Var}_{\mu,X}(z_{IN}) \approx \mathrm{Var}_{\mu,X}(z_{OUT}) = \frac{d}{T}\left[ \bar{\sigma}^4 + \frac{\sigma^4}{kN} + \left( \frac{k+N}{kN} \right)(\bar{\sigma}^2 \sigma^2) \right]$$

We note that the **strong** adversary always achieves greater attack success rates than the **weak** adversary because the distance between the expected value of the test statistic, $z$, when the task is **IN** or **OUT** is strictly larger than that of the **weak** adversary at a similar level of variance. Furthermore, this attack on mean estimation shows that the adversary's success is dependent on two distinct parts: (1) the knowledge they receive from having *member* data (i.e. the $\frac{d}{TN}\sigma^2$ term) and (2) the knowledge they have about the *distributions*, or *tasks*, that compose the dataset (i.e. the $\frac{d}{T}\bar{\sigma}^2$ term).

## 5 METHODOLOGY

In this section, we realize the threat model in Section 3 for machine learning models by introducing two efficient, purely black box attacks on shared representations.

### 5.1 OUR TASK-INFERENCE ATTACKS

Our two simple and efficient black-box attacks are inspired by prior work on membership-inference, which shows that embeddings of training samples are more robust to augmentations (e.g., random rotations) than non-members (Liu et al., 2021a). Our key insight is to generalize this principle by treating distinct samples from the same task as "natural" augmentations of one another. We therefore hypothesize that their embeddings will exhibit strong codependencies. We argue this is caused by distribution-level memorization, where the shared representation "overfits" to implicitly learned properties of entire task distributions, irrespective of the training data's labeling in MTL, in a manner analogous to sample-level memorization.

**Coordinate-Wise Variance Attack (Algorithm 1)**   To mount the *coordinate-wise variance attack*, the adversary first queries the shared representation on a batch of $k$ data samples from a given task, then aggregates the embeddings into a set $E = \{h_\theta(x_1), \ldots, h_\theta(x_k)\}$. Then, the adversary computes the empirical covariance matrix of $E$ and takes the trace divided by the dimension of the embedding vectors to be the task-inference statistic, $z$. This is equivalent to computing the sum of coordinate-wise variances of the embeddings. Lastly, the adversary sets a threshold $\gamma$ such that any task with $z > \gamma$ is labeled as $IN$, and any task with $z < \gamma$ is labeled as $OUT$. The full attack algorithm can be found in Appendix A.1.

**Pairwise Inner Product Attack (Algorithm 2)**   For our second attack, we once again use the fact that the shared representation produces close embeddings for data from the same task, but we measure similarity of the entire embedding vectors rather their individual coordinates by taking their *inner products* (or *cosine similarities*). As in the previous attack, the adversary first queries the shared representation and aggregates the embeddings $E$. Then, for each unique pair of data samples $(x_i, x_j)\ i \neq j$, the adversary computes the absolute value of the inner product of their embeddings, $z$, and stores the value in a set $S$. The adversary then takes the mean of the $z$'s ($\bar{S}$) and applies a threshold $\gamma$ as in the variance attack. The full version of Algorithm 2 can be found in Appendix A.1.

**Normalizing Embeddings**   Unlike prior work on membership-inference (Shokri et al., 2016; Carlini et al., 2022) and property inference (Chaudhari et al., 2022; Mahloujifar et al., 2022) attacks that train shadow models, our task-inference adversary requires only query access to the shared representation, eliminating the need for shadow, or reference, models to calibrate the attack. While a slightly stronger adversary could construct a labeled, auxiliary dataset of known **OUT** tasks to calibrate their attack and perform a more powerful exact one-sided test, as in (Carlini et al., 2022), we find that simple thresholding is sufficient to achieve high success rates while maintaining the purely black-box aspect of the threat model. Thus, using the adversary's sampling access to task data and query access to the shared representation, we attempt to reduce the noise in embedding vectors by applying a *whitening transformation* (A.7.1).

## 6 EVALUATION

We present a thorough evaluation of our efficient, black-box task-inference attacks on MTL models across the vision and language domains. We study the performance for both **strong** and **weak**

adversaries, and we investigate the sources of task-inference leakage by analyzing how attack success correlates with the model's generalization. We note the practicality of our attacks, requiring as few as four samples and taking roughly *0.1 seconds* per black-box query and task-inference prediction on a single RTX 4090 GPU. Additional details on these experiments are available in Appendix A.

## 6.1 MTL Training

In this study, we consider the original instantiation of MTL training which is described in the seminal work on the topic of MTL (Caruana, 1997). We highlight that this MTL algorithm is a slight variation of centralized FedSGD (McMahan et al., 2017), one of the baseline algorithms in the original paper on collaborative learning. For all of our models, we allow all tasks to share all but the task-specific classification heads. In our experiments, the shared representation takes the form of a neural network, and the classification heads are linear. During MTL training, we perform a forward pass and route each embedding vector to its corresponding task-specific layer. Thus, in the backward pass, the loss from *all* tasks is used to train the shared encoder, while only the loss from each particular task is used to train each linear layer. A detailed description of the multitask training setup used in our evaluation, along with the corresponding algorithm (Algorithm 3) can be found in Appendix A.3.

## 6.2 Models, Datasets, and Metrics

In our vision experiments, we use ResNet models (He et al., 2015) of differing sizes as the shared representation for MTL. We evaluate our attacks on these vision models using the CelebA (Liu et al., 2015) and Federated EMNIST (FEMNIST) (Caldas et al., 2019) datasets. A detailed description of our FEMNIST experiments can be found in Appendix A.6. For our language experiments, we use downsized variants of the BERT (Devlin et al., 2018; Turc et al., 2019; Bhargava et al., 2021) architecture, which are pretrained on MNLI (Williams et al., 2018), for downstream classification. We additionally evaluate our attack on Gemma 3 270M (Team et al., 2025) fine-tuned for personalized next-token prediction on the Reddit dataset (Völske et al., 2017).

To measure the performance of our task-inference attacks, we use metrics that are commonly found in the inference attack literature (Carlini et al., 2022; Zarifzadeh et al., 2024; Ye et al., 2022; Kandpal et al., 2023). We report the ROC curves of our attack, along with true positive rates (TPR) at fixed low false positive rates (FPR) and area under the curve (AUC). To highlight the black-box nature of our attack, report the (TPR, FPR) pairs when blindly thresholding our test statistics at the 50th, 75th, and 90th percentiles. More details on the models, datasets, and metrics used in this evaluation can be found in Appendices A.2 and A.4.

## 6.3 Personalization

Here, we present the results of our experiments in the vision and language settings where MTL is used for *personalization*. Across all experiments, the datasets are organized by individual users, with each user contributing multiple samples. In the MTL framework, the multiple tasks are the users, and each user's task-specific layer, which is linear in this case, receives updates from their data. The shared representation is trained on all of the tasks. When the adversary infers a task's inclusion in multitask training, they are leaking whether or not an individual's data was present at all in updates to the shared representation. Like in membership-inference, the **strong** adversary will have a batch of real training samples at their disposal to perform this test, but the **weak** adversary receives fresh samples that belong to the user and were never seen during training. Additional details for all of our experiments, along with additional results for the FEMNIST dataset, can be found in Appendix A.5.1.

**CelebA** First, we report the results for our attacks in the vision setting. We train a ResNet-34 (He et al., 2015) model, pre-trained on ImageNet (Deng et al., 2009), to perform binary facial attribute detection on the CelebA dataset. In this MTL setup, each task corresponds to a unique person, and we train a separate linear layer for each of the 512 total tasks (256 IN, 256 OUT).

The results of this experiment are shown in Figure 1a and Table 1. We find that while both the **strong** and **weak** adversaries can achieve non-trivial success rates in determining a task's inclusion in training, the true positive rate of the **weak** adversary is bounded above by the true positive rate of the **strong** adversary for any fixed FPR. Moreover, we see that our variance attack nets better TPR in the

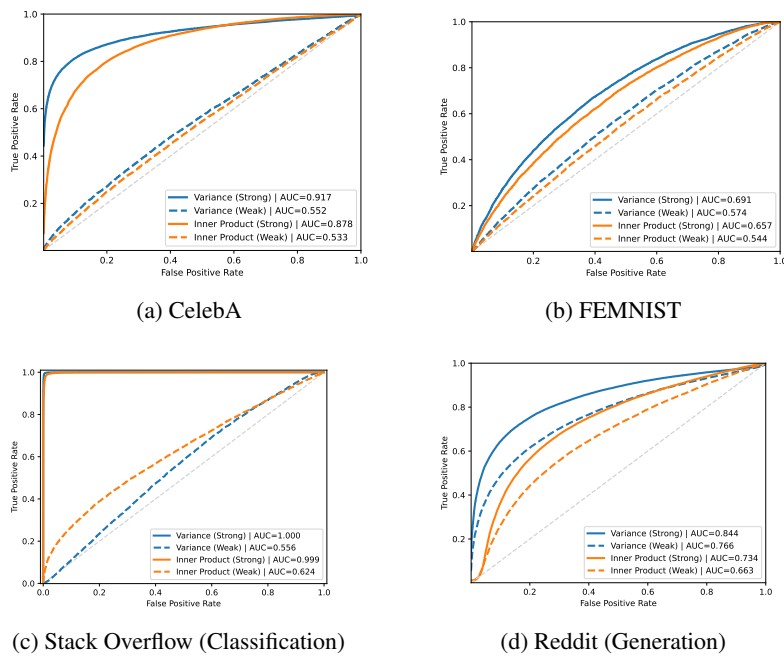

Figure 1: ROC curves of our Task-Inference Attacks (MTL for Personalization)

low FPR regime, but the inner product attack sees better FPR at higher FPR. At a fixed FPR of $1\%$, the TPR of our variance attack on CelebA achieves a TPR of $61.2\%$ and $2.9\%$ for the **strong** and **weak** adversaries, respectively.

**Stack Overflow** Next, we present the results for our language experiments on the StackOverflow dataset. We train a BERT Small (Turc et al., 2019) model to perform topic classification, where the tasks are users who contributed posts to Stack Overflow, with 256 tasks total (128 IN, 128 OUT).

Figure 1c and Table 1 show the results for this attack. We find that the **strong** adversary achieves nearly perfect AUC for both attacks and an empirically-observed FPR of 0% at the $75^{th}$ and $90^{th}$ percentile thresholds over all runs of the attack. The **weak** adversary again sees nontrivial AUC for both attacks, and we see a tradeoff between the TPR of the variance and inner product attacks at particularly high FPR. At fixed FPR of 1% the inner product adversary with **weak** access to the data achieves a TPR of 8.2%, while the **strong** adversary achieves a TPR of 98.5%. By analyzing the topic frequencies in the dataset, we empirically find that individual users tend to only write about a small subset of topics. Across all posts, the median user posts about 31 (or 12.1%) of the 256 total topics, which could lead to high distinguishability of users in representation space.

**Reddit** Lastly, we present the results for our language experiments on the Reddit dataset (Völske et al., 2017). We fine-tune a pretrained Gemma 3 (Team et al., 2025) model for next-token prediction, where the 256 IN tasks are Reddit users with 64 to 512 posts each, and we evaluate our attack over 512 total tasks (256 IN, 256 OUT). To personalize the model, we train a task-specific LoRA Hu et al. (2022) adapter on the language modeling head for each user, while all tasks jointly update the rest of the parameters.

Figure 1d and Table 1 show the results for this experiment. We find that both the **strong** and weak adversaries achieve high attack AUC with both the inner product and variance attacks. At a fixed FPR of 1%, the variance attack achieves a TPR of 19.7% and 19.0% for the **strong** and **weak** adversaries, respectively. We observe that the coordinate-wise variance attack consistently outperforms the inner product attack in this setting because (1) the low-rank adapters used for personalization induce a spread in the embedding space for trained users and (2) the learned embeddings are not intended for classification, thus they do not exhibit the same clustering properties we observed in our experiments on discriminative models.

### 6.4 MULTIPLE LEARNING PROBLEMS

We present the results of our experiments on models trained using MTL to solve multiple related learning problems. In contrast to the datasets used to train MTL models in the experiments presented from Section 6.3, the datasets are split by learnable classes in the dataset rather than by person. For example, Stack Overflow can be split into its constituent topics, and we can have an MTL model with a tailored head for detecting each individual topic's presence. We draw attention to the fact that in this setting, the labels for the learning problem are directly tied to the task. In other words, if a task is not included, there are no positive labels corresponding to that task in the training dataset. Additional details for the experiments in this section can be found in Appendix A.5.2.

**CelebA**    We use the ResNet50 (He et al., 2015) architecture as the backbone for our MTL model trained on CelebA as there is ample data per task. Each of the 20 IN tasks (40 total) has a corresponding linear head for binary attribute prediction, and no two tasks share the same labeling. For example, the task head dedicated to hair color does not make predictions for the task head dedicated to detecting glasses.

In Table 2 and Figure 2a, we see that the **strong** adversary is able to achieve an AUC of $0.745$ using our coordinate-wise variance attack and a TPR of 22.8% at a FPR of 0% over all runs when using the $90^{th}$ percentile threshold. The **weak** sees nontrivial success rates, with a TPR roughly $2\times$ larger than the FPR when using the $90^{th}$ percentile threshold.

**Stack Overflow**    In this experiment, each of the 140 total tasks corresponds to the inclusion of a positively labeled topic in the dataset. For example, if the Stack Overflow topic "Python" is **IN**, there is a task head dedicated to detecting whether a post includes the topic "Python". Using a BERT Medium (Turc et al., 2019) (41M parameters) model as the shared representation, we use MTL to learn linear task heads that are specialized to detecting the presence of a single topic within a post.

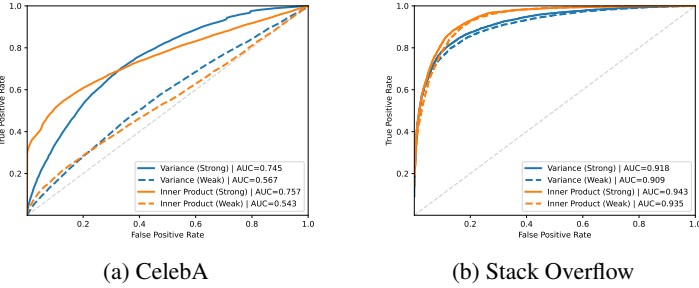

(a) CelebA                              (b) Stack Overflow

Figure 2: ROC of Task-Inference Attacks (MTL for Multiple Learning Problems)

Figure 2b shows that in the setting where tasks correspond directly to the training data labeling in MTL, the gap in attack performance between the **strong** and **weak** adversaries becomes small. This can be attributed to the fact that the model necessarily has to achieve high utility on the training tasks in order to solve the learning problems. This contrasts our findings in Section 6.3, where the task heads in MTL are not solving distinct learning problems, and the separation between both adversaries is notably larger. In Table 2, we see that the (TPR, FPR) pairs for both adversaries are nearly equal, and the **weak** adversary is able to achieve a 19.6% TPR at 0.2% FPR.

### 6.5 INVESTIGATING SOURCES OF TASK-INFERENCE LEAKAGE

We investigate the factors that lead to task-inference leakage by studying how attack success varies with the model's generalization gap. Theorem 4.1 shows an advantage for the **strong** adversary, and we observe the gap between **strong** and **weak** adversaries is large in our MTL experiments, especially with sparse tasks like in CelebA. Similar to membership-inference, task-inference success is closely tied to the model's generalization gap. To explore this, we measured the inner product attack's AUC on Stack Overflow models (trained for personalization as in Section 6.3) at different stages of training.

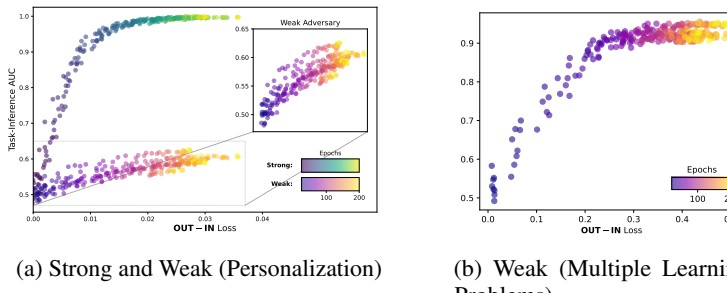

(a) Strong and Weak (Personalization)    (b) Weak (Multiple Learning Problems)

Figure 3: Relationship between Generalization Gaps and Inner Product Attack AUC on Stack Overflow

As shown in Figure 3a, the **strong** adversary's AUC rapidly increases as the generalization gap, or difference between training loss and a validation loss on OUT task data, grows. This finding mirrors traditional membership-inference, where leakage is linked to overfitting on particular training examples. In contrast, the **weak** adversary's AUC grows with a different quantity: the gap between the loss on *unseen samples from IN tasks* and the validation loss on OUT samples. This suggests the shared representation is not only memorizing training points but also generalizing more effectively on the training tasks. The observation that models memorize information about tasks as a whole is consistent with findings from prior work on user-inference attacks on LLMs (Kandpal et al., 2023).

## 7  CONCLUSION AND DISCUSSION

In this work, we study privacy leakage when learning models over a mixture of tasks in multitask learning. In particular, we focus on the setting where MTL is used to jointly train a model for many tasks at once by learning a shared representation that captures common features between tasks. We propose a novel, purely black-box *task-inference* threat model, where the adversary's goal is to infer the inclusion of a target task in training given only query access to the smallest shared component when learning jointly over many tasks, the shared representation. By analyzing our task-inference threat model in the context of tracing attacks on Gaussian mean estimation, we find a separation between **strong** adversaries with access to training samples and **weak** adversaries with access to fresh samples from the target task's distribution that did not appear in the training dataset. To verify our analysis, we propose purely black-box attacks on machine learning models trained with MTL and we conduct extensive experimentation in the vision and language domains for multiple use cases of MTL. We find that our attacks can consistently achieve non-trivial success rates in terms of AUC and TPR, even when the adversary has no reference knowledge and chooses thresholds based on percentiles. Additionally, we attempt to understand the factors that lead to task-inference leakage by measuring how attack AUC varies for both the **strong** and **weak** adversaries as a function of the model's generalization gap.

## 8  ACKNOWLEDGMENTS

We are grateful to the ICLR reviewers for useful feedback and comments. Adam Smith was supported at BU by US NSF award CNS-2232694. Part of this work was done while he was a visiting scientist at Google Deepmind. John Abascal and Alina Oprea were supported at Northeastern by US NSF award CNS-2247484 and by a grant from Coefficient Giving. Jonathan Ullman was supported at Northeastern by US NSF awards CNS-2232694 and CNS-2247484.

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

## A  ADDITIONAL RESULTS AND DETAILS FROM METHODOLOGY AND EVALUATION

In this section, we provide additional details, figures, and results for Sections 5 and 6

## A.1 ATTACK ALGORITHMS

We provide full versions of the attack algorithms described in Section 5.1.

---
**Algorithm 1** Coordinate-Wise Variance Attack
---
1: **Input:** A shared representation $h_\theta$, a challenge set $X_\tau$
2: $E = \{\}$
3: **for** $x_i \in X_\tau$ **do**
4:    $e_i \leftarrow h_\theta(x_i)$ and $E \leftarrow E \cup \{e_i\}$       ▷ Query rep. on challenge set
5: Compute $Q$, the empirical covariance matrix of $E$
6: **return** $tr(Q)$       ▷ Return avg. coordinate-wise variance

---
**Algorithm 2** Pairwise Inner Product Attack
---
1: **Input:** A shared representation $h_\theta$, a challenge set $X_\tau$
2: $E = \{\}$
3: **for** $x_i \in X_\tau$ **do**
4:    $e_i \leftarrow h_\theta(x_i)$       ▷ Query rep. on challenge set
5:    $e_i \leftarrow W(e_i - \bar{e})$       ▷ Apply whitening
6:    $E \leftarrow E \cup \{e_i\}$       ▷ Store normalized embedding
7: $S = \{\}$
8: **for** all unique pairs $(e_i, e_j)$; $i \neq j$; $e_i, e_j \in E$ **do**
9:    **if** use_cosine_similarity **then**
10:       $(e_i, e_j) \leftarrow \left( \frac{e_i}{\|e_i\|}, \ \frac{e_j}{\|e_j\|} \right)$
11:    $z \leftarrow |\langle e_i \ , \ e_j \rangle|$ and $S \leftarrow S \cup \{z\}$
12: **return** $\bar{S}$       ▷ Return avg. pairwise inner product

---

## A.2 MODELS AND DATASETS

### A.2.1 VISION MODELS

In our vision experiments, we use ResNet models He et al. (2015) of differing sizes as the shared representation for MTL. The ResNet architecture has gained widespread adoption across computer vision applications due to its computational efficiency and high performance on a variety of datasets. This model architecture makes use of residual connections, which stabilize training and convergence to help maintain high utility when trained on large image datasets. When available, such as in our CelebA experiments, we use larger ResNet models which are pretrained on the ImageNet Deng et al. (2009) dataset from the PyTorch Paszke et al. (2019) library.

### A.2.2 LANGUAGE MODELS

For our language experiments, we use the BERT Devlin et al. (2018) architecture as the shared representation. In particular, we use the downsized variants of BERT Turc et al. (2019); Bhargava et al. (2021) which are pretrained on MNLI Williams et al. (2018) for downstream sequence classification.

### A.2.3 VISION DATASETS

We evaluate our task-inference attacks on both the CelebA Liu et al. (2015) faces dataset and the Federated EMNIST (FEMNIST) Caldas et al. (2019) handwritten character and images dataset. CelebA contains high-resolution images of celebrities, each with a unique identifier, and 40 facial attributes with binary labels. The FEMNIST dataset contains 28x28 grayscale images of 62 different types of handwritten characters. In our experiments on CelebA, we split the dataset into tasks by person and by facial attribute for each of the two MTL use cases we consider. We split FEMNIST by writer and train an MTL model for character recognition, personalized to each writer's hand-drawn images.

### A.2.4  Language Datasets

To evaluate our attack in the language setting, we use the Stack Overflow Annamoradnejad et al. (2022) dataset, which consists of posts from the Stack Overflow website with corresponding ratings, user ID, and topic tags for each post. In our evaluation, we split the posts into tasks by user ID and by topic for each of the MTL use cases.

### A.3  Multitask Training

Algorithm 3 shows the general training loop we use for MTL. In practice, we use AdamW Loshchilov & Hutter (2017) for our `Update` step, and we perform shuffling over tasks and batching for efficiency. Additionally, when there is sufficient data available per task, we randomly subsample data from each task for each training epoch.

---

**Algorithm 3** Multitask Learning Training Loop

1: **Input:** A shared representation $h_\theta$, $T$ task-specific layers $\{g_{\beta_1}, \ldots, g_{\beta_T}\}$, a training set split by task $D = \{(X_1, y_1) \ldots (X_T, y_T)\}$, number of training rounds $t$, a loss function $\mathcal{L}$
2: **for** $i \in [t]$ **do**
3:    $\ell_{MTL} \leftarrow 0$                                                        ▷ Initialize multitask loss
4:    **for** $j \in [T]$ **do**
5:        $p_j \leftarrow g_{\beta_j}(h_\theta(X_j))$                                   ▷ Get predictions for task $j$
6:        $\ell_{MTL} \leftarrow \ell_{MTL} + \frac{1}{T}\mathcal{L}(p_j, y_j)$
7:    **for** $j \in [T]$ **do**                                                       ▷ Update each task specific layer
8:        $\beta_j \leftarrow \texttt{Update}\left(\beta_j, \nabla_{\beta_j}\ell\right)$
9:    $\theta \leftarrow \texttt{Update}(\theta, \nabla_\theta\ell)$                   ▷ Update shared representation
    **return** $h_\theta$

---

Since the number of samples per task can be smaller than the embedding dimension, we use several regularization techniques to keep the task-specific linear layers from overfitting to noisy embeddings early during MTL training. To summarize, we use large values for weight decay, or $L_2$ regularization, learn a bottleneck on the shared representation to decrease the embedding dimension, normalize gradients Chen et al. (2018) across tasks during model updates, apply clipping to gradients, and, in our vision experiments, apply standard augmentations to training images.

In our vision experiments, the CelebA dataset has very few samples per task-specific layer; 22 per celebrity after holding out samples for the weak adversary. Thus, we do two iterations of "warm start" training Nguyen et al. (2023), where we freeze the shared representation and optimize the task-specific layers for 40 epochs with with a low learning rate (e.g. $\eta = 10^{-4}$). In our language experiments where we use BERT models, we do 20 epochs of warm start training. We also apply a linear projection on the typical, high-dimensional penultimate layer of our BERT and ResNet models to 16 or 32 dimensions in order to reduce the chance of task-specific layers overfitting. Each training step in MTL is performed over a minibatch of tasks, rather than the entire accumulated gradient of the dataset. We additionally use gradient clipping ($C = 1$), gradient normalization across tasks Chen et al. (2018), and a smaller weight decay parameter (e.g. $\lambda = 10^{-4}$) than the task-specific layers (e.g. $\lambda = 10^{-3}$) to regularize the shared layers. In total, we train for 200-300 epochs, or communication rounds, for all of our models, passing through all of the tasks in the dataset each time.

### A.4  Metrics

Throughout our evaluation, we analyze the ROC curves of our attack, which measure the relationship between true positive rate (TPR) and false positive rate (FPR). As summary statistics, we report the area under the ROC curve, or AUC, as well as the TPR at fixed low FPR (e.g. 1%). Our evaluation distinguishes between two adversarial settings: a **strong** adversary with access to training samples and a **weak** adversary limited to auxiliary data from the challenge task that was never seen during training. Thus, we present ROC curves for each adversary. Additionally, we highlight that our attacks can be performed in a purely black-box setting. This results in the adversary not necessarily having sufficient knowledge to select the threshold that yields an optimal tradeoff between TPR and FPR.

So, we also report the (TPR, FPR) pairs when thresholding our test statistics at the 50th, 75th, and 90th percentiles.

## A.5 EXPERIMENTAL SETUP

Here, we provide details for the experimental setup for both of the MTL use cases we study. In each of our experiments, we aggregate results over several training runs and many attack trials per training run. In each trial, we subsample the task-specific data such that the adversary has a fresh combination of few samples from each of the target tasks.

### A.5.1 PERSONALIZATION

**CelebA**   We filter CelebA such that each unique individual has roughly 30 images containing their face, and we hold out 8 samples per individual to mount an attack with the **weak** adversary. In our experiments, we jointly train the MTL model on 256 total tasks with 22 samples per task, or roughly 5600 total samples, and thus use several regularization techniques to ensure that the ResNet layers learn performant embeddings. During each training step, we randomly sample a batch of tasks and update their corresponding task heads, while aggregating these tasks' gradients to update the shared representation. Once the model is trained, we run 128 trials of each attack on all 512 tasks (256 $IN$ and 256 $OUT$), using 8 samples per task for the **strong** adversary and 4 of the held out samples per task for the **weak** adversary.

**Stack Overflow**   We filter the dataset such that each individual has at least 48 total posts; 32 posts for the model's training set and 16 to be used by the **weak** adversary. Then, we split the dataset into 128 **IN** tasks and 128 **OUT** tasks, yielding two datasets of roughly 10k posts, with each user contributing about 140 posts on average. We train the shared representation and task-specific linear layers jointly on the **IN** dataset. The MTL model is trained to detect the presence of 256 unique topics in posts, and each post can have multiple corresponding topics. After training is complete, we run each attack on all 256 tasks for 128 trials, using 8 samples and 4 samples per trial for the **strong** and **weak** adversary, respectively.

**Reddit**   We filter the Reddit dataset such that each unique user (or task) has at least 64 posts and at most 512 posts. We set aside 16 posts per user for the **weak** adversary. Then, we split the users into 256 **IN** tasks and 256 **OUT** tasks. We start with a pretrained Gemma 3 270M model and jointly train a shared representation and task-specific LoRA adapters (with $r = 16$ and $\alpha = 32$) for next-token prediction. After each training run is complete, we run each attack on all 512 tasks for 256 trials, subsampling 16 samples and 8 samples per trial for the **strong** and **weak** adversary, respectively.

### A.5.2 MULTIPLE LEARNING PROBLEMS

**CelebA**   For our evaluation on CelebA when MTL is used to solve multiple learning problems simultaneously, we split the dataset into potentially overlapping tasks by facial attribute, 20 **IN** and 20 **OUT**, with at least 1024 samples with corresponding positive and negative labelings for each task. Because some tasks have very low positive label frequencies (e.g. $< 1\%$), this minimum sample size ensures sufficient positive examples for the task head to learn from. We average the results of our experiments over 8 MTL runs, and run the attacks on the 40 total tasks 128 times each, using 16 samples and 8 samples for the **strong** and **weak** adversary, respectively.

**Stack Overflow**   In these experiments on Stack Overflow, in contract to the experiments in Section 6.3, the posts are not split by user. Rather, each of the 140 total tasks corresponds to the inclusion of a positively labeled topic in the dataset. If a particular topic is *not* included, there is no task head that can learn positively labeled "Python" samples. Additionally, because the posts in the Stack Overflow dataset can contain multiple topics, the training data is not disjoint between tasks; the labeling of each task dataset is unique. We randomly split the data into 70 **IN** tasks and 70 **OUT** tasks, with each task-specific dataset containing 1024 posts. We run both of our attacks on the BERT model 128 times, using 32 samples and 16 samples each trial for the **strong** and **weak** adversaries, respectively.

Table 1: (TPR, FPR) and Balanced Accuracy for Black-Box Percentile Thresholds for Inner Product Attack at $50^{th}$, $75^{th}$, and $90^{th}$ Percentile Thresholds (Personalization)

| Dataset | Access | $50^{th}$ Percentile (TPR, FPR) | Acc. | $75^{th}$ Percentile (TPR, FPR) | Acc. | $90^{th}$ Percentile (TPR, FPR) | Acc. |
|---|---|---|---|---|---|---|---|
| CelebA (Fig 1a) | **Strong** | $(80\%, 20\%)$ | 80% | $(46.7\%, 3.3\%)$ | 71.7% | $(19.5\%, 0.5\%)$ | 59.5% |
| | **Weak** | $(52.4\%, 47.6\%)$ | 52.4% | $(27.4\%, 22.5\%)$ | 52.5% | $(11.5\%, 8.5\%)$ | 51.6% |
| FEMNIST (Fig 1b) | **Strong** | $(61.1\%, 38.8\%)$ | 61.1% | $(33.5\%, 16.4\%)$ | 58.7% | $(14.3\%, 5.7\%)$ | 54.3% |
| | **Weak** | $(53.2\%, 46.8\%)$ | 53.2% | $(27.2\%, 22.8\%)$ | 52.2%, | $(11.2\%, 8.8\%)$ | 51.2% |
| Stack Overflow (Fig 1c) | **Strong** | $(98.7\%, 1.2\%)$ | 98.7% | $(50.0\%, 0\%)$ | 75.0% | $(19.9\%, 0\%)$ | 59.9% |
| | **Weak** | $(58.2\%, 41.7\%)$ | 58.2% | $(34.1\%, 15.8\%)$ | 59.1% | $(16.3\%, 3.6\%)$ | 56.3% |
| Reddit (Fig 1d) | **Strong** | $(68.7\%, 31.3\%)$ | 68.7% | $(39.1\%, 10.9\%)$ | 64.1% | $(14.8\%, 5.2\%)$ | 54.8% |
| | **Weak** | $(62.6\%, 37.4\%)$ | 62.6% | $(35.5\%, 14.5\%)$ | 60.5% | $(14.1\%, 5.9\%)$ | 54.1% |

Table 2: (TPR, FPR) and Balanced Accuracy for Black-Box Percentile Thresholds for Inner Product Attack at $50^{th}$, $75^{th}$, and $90^{th}$ Percentile Thresholds (Multiple Learning Problems)

| Dataset | Access | $50^{th}$ Percentile (TPR, FPR) | Acc. | $75^{th}$ Percentile (TPR, FPR) | Acc. | $90^{th}$ Percentile (TPR, FPR) | Acc. |
|---|---|---|---|---|---|---|---|
| CelebA (Fig 2a) | **Strong** | $(68.9\%, 31.1\%)$ | 68.9% | $(42.6\%, 7.4\%)$ | 67.6% | $(22.8\%, 0\%)$ | 61.4% |
| | **Weak** | $(52.5\%, 47.5\%)$ | 52.5% | $(29.0\%, 21.0\%)$ | 54% | $(13.5\%, 6.6\%)$ | 53.5% |
| Stack Overflow (Fig 2b) | **Strong** | $(87\%, 13.9\%)$ | 87% | $(48.3\%, 1.7\%)$ | 73.3% | $(19.9\%, 0.1\%)$ | 59.9% |
| | **Weak** | $(85.9\%, 14.1\%)$ | 85.9% | $(47.6\%, 2.4\%)$ | 72.6% | $(19.6\%, 0.2\%)$ | 59.7% |

## A.6 FEMNIST RESULTS

We report additional results in the vision setting for Federated EMNIST Caldas et al. (2019). We train a MTL model with ResNet8 He et al. (2015) as the shared representation and personalize each task-specific layer to a unique writer in the dataset. Each writer in the FEMNIST has significantly more data than each individual in CelebA. Thus, we train the MTL model on 128 tasks with 128 samples per task and hold out 16 samples to mount our attack with the **weak** adversary. For each of the 128 runs of the attack, the **strong** adversary receives 16 training samples, and the **weak** adversary receives 8 of the held out samples. Figure 1b and Table 1 show the ROC curves and (TPR, FPR) pairs of our task-inference attacks on Federated EMNIST. We observe that both the **strong** and **weak** adversaries are able to achieve nontrivial success rates, and the **strong** adversary sees a TPR of 3% at a fixed 1% FPR. When the **weak** adversary picks the $90^{th}$ percentile threshold, the inner product attack yields a $2.5\times$ higher TPR than FPR.

## A.7 TABLES FOR PERCENTILE-BASED THRESHOLDS

Here, we present the tables which contain the results of our experiments where we choose the decision threshold for our test statistic using the $50^{th}$, $75^{th}$, and $90^{th}$ percentile values.

### A.7.1 WHITENING TRANSFORMATION

Applying a whitening transformation to the embeddings transforms them into random vectors with covariance equal to the identity matrix. By doing this, we attempt to contract the axes in the the representation space that dominate our task-inference statistics. We find that whitened embeddings often provide better signal to the adversary for our inner product attack (Algorithm 2) than raw embeddings. To compute the transformation, we first estimate the covariance matrix of the embedding space by pooling all of the embeddings available to the adversary, regardless of task or inclusion in the model's training set, and computing the regularized covariance matrix $\Sigma_{reg} = (Q + \lambda \cdot \frac{tr(Q)}{d} \mathbb{I}_d)$ where $Q$ is the sample covariance matrix, $d$ is the embedding dimension, and $\lambda$ is a small constant. Once a well-conditioned $\Sigma_{reg}$ has been estimated, a common choice for the whitening transformation would be $W = \Sigma_{reg}^{-1/2}$. To ensure that the covariance estimate is not dependent on a challenge task's data, we compute whitening transformations for each of the $2T$ tasks that we input to our attack, leaving out the data from one task each time. When applying $W$ to the embeddings, we additionally center the embeddings by computing $\bar{e}$, the sample mean of *all* embedding vectors. Thus, we normalize a given embedding $e$ by computing $W(e - \bar{e})$.

### A.7.2 Variance Attack Test Statistic

Across all of our experiments in both MTL settings, the statistic produced by our inner product attack (Algorithm 2) is consistent with respect to the ordering of **IN** and **OUT** distributions. In contrast, when studying leakage in MTL for personalization, we observe a discrepancy in the test statistic distributions for vision and language datasets. Figure 4 shows that the coordinate-wise variance statistic (Algorithm 1) is larger for **IN** tasks than **OUT** tasks.

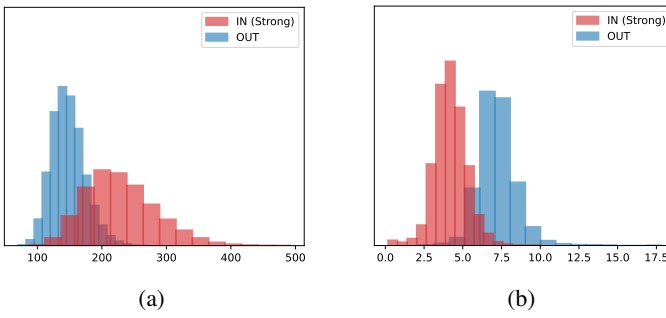

(a)                                    (b)

Figure 4: Distribution of Coordinate-Wise Variance Statistic; (a) CelebA; (b) Stack Overflow

To investigate this discrepancy in the ordering of our coordinate-wise variance test statistic, we compute the pairwise inner product, or similarity, of the weights in the task-specific layers for our CelebA, FEMNIST, and Stack Overflow models after MTL training. Figure 5 shows the distributions of the task head inner products, where we see that the vision models produce embeddings that lead task-specific layers to be correlated on average (i.e. inner product not centered around 0). Because the task heads are correlated, the primary signal for our attack is the shared representation's "overconfidence" in certain directions of the embedding space, which yields higher coordinate-wise variance. We find that the task heads of our FEMNIST models have the highest correlation, followed by our CelebA models, then Stack Overflow, which maps to the ordering of the AUC scores we see in our experiments on MTL for personalization.

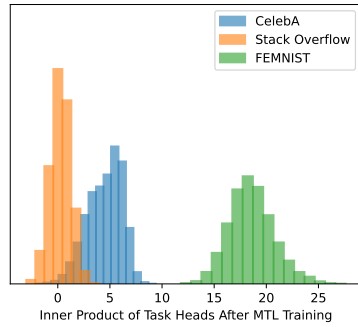

Figure 5: Task-Specific Layer Similarity (Personalization)

### A.8 Ablations on Synthetic Data

We attempt to understand task-inference leakage by running our attacks on a synthetic dataset. The data generation process for this dataset is identical to the mean estimation example in Section 4, but we adapt it for machine learning. To create the synthetic dataset, we start by sampling $2T$ i.i.d. tasks, $\{\mu_1 \ldots \mu_{2T}\}$ from a $d$-dimensional Gaussian distribution. For each of the tasks, $\mu_i$, we sample a dataset of $N$ $d$-dimensional vectors from the corresponding task distribution. To label the data for a learning problem, we first generate a random projection matrix $H \in \mathbb{R}^{k \times d}$, where $d$ is the data dimension and $k$ is the embedding dimension. Then, we randomly sample the "target" task-specific layers $g_1, \ldots g_T \in \mathbb{R}^k$ and get the label for each sample, $\vec{x}$, coming from task $\tau \in [T]$ to be the sign of the inner product between $g_\tau$ and the embedding vector $H\vec{x}$ (that is, for any sample $X_{i,j}$,

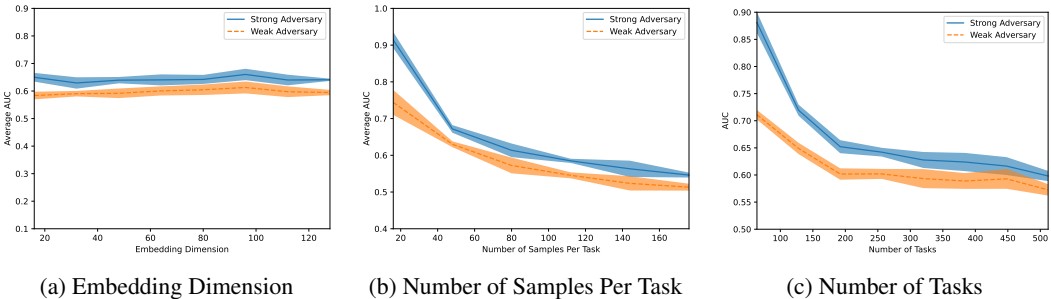

(a) Embedding Dimension      (b) Number of Samples Per Task      (c) Number of Tasks

Figure 6: Ablations on Synthetic Data

$y_{i,j} = \texttt{sign}(\langle\, g_i\,,\, HX_{i,j}\,\rangle)$. This particular dataset is well suited for MTL as all tasks have unique labeling functions, but a common projection into the embedding space.

In our experiments on synthetic data, we use a simple neural network with 512 hidden units and a linear projection layer into the embedding space to approximate $H$. We vary the embedding dimension, number of samples per task, and number of tasks in the dataset, then measure our coordinate-wise variance attack's AUC over 4 training runs, with 64 trials of each attack on 8 random samples from each task. The results of this experiment are shown in Figure 6. We observe that increasing the number of total samples in the dataset, whether by increasing the number of samples per task or the total number of tasks, has a sharp impact on the **strong** adversary's AUC. As the model has more samples to learn from, the gap between the **strong** and **weak** adversary's AUC scores shrinks. We also find that, in this suite of experiments, the embedding dimension has little impact on task-inference AUC.

## B   ANALYZING TASK-INFERENCE VIA TRACING ATTACKS

To motivate further exploration of attacks on deep learning models, we present a simplified analog of multitask learning via mean estimation. Proofs of our theorems in this Section can be found in Appendix B.2.

Suppose that there exists a set of *task means* $M = \{\mu_1, \ldots, \mu_T\}$ which are sampled i.i.d. from $\mathcal{N}(\bar{\mu}, \bar{\sigma}^2 \mathbb{I}_d)$ where $\bar{\mu}$ is the "true" mean that we would like to estimate. For each "task" $\mu_i$, we can sample $N$ (where $N$ is relatively small) points i.i.d. from $\mathcal{N}(\mu_i, \sigma^2 \mathbb{I}_d)$ and store them in the set $X_i = \{x_{i,1}, \ldots, x_{i,N}\}$. Since no individual dataset $X_i$ would yield an accurate estimate of $\bar{\mu}$ we can instead compute the sample "multitask" mean using all of the $X_i$'s as

$$\hat{\mu} = \frac{1}{T} \sum_{i=1}^{T} \left( \frac{1}{N} \sum_{j=1}^{N} X_{i,j} \right)$$

This "multitask" mean can be viewed as analogous to the weights of the shared representation in multitask learning, because we average over data, $X_i$, sampled from a mixture of Gaussian distributions, $\mathcal{N}(\mu_i, \sigma^2 \mathbb{I}_d)$, which share features, to produce an accurate estimate of the underlying common parameter, $\bar{\mu}$. We note that $\hat{\mu}$ is Gaussian with expectation and covariance

$$\mathbb{E}_{\mu, X}[\hat{\mu}] = \bar{\mu} \qquad \mathrm{Cov}(\hat{\mu}) = \left( \frac{\bar{\sigma}^2}{T} + \frac{\sigma^2}{N \cdot T} \right) \cdot \mathbb{I}_d$$

### B.1   TRACING ATTACK FOR TASK INFERENCE

Here, we construct an adversary, based on prior work on tracing Dwork et al. (2015), with similar variants to the **strong** and **weak** adversaries detailed in Section 3. We consider a challenger who releases the statistic $\hat{\mu}$ and an adversary who wants to learn whether a given task $\mu_i$ was included in the dataset that was used to compute $\hat{\mu}$.

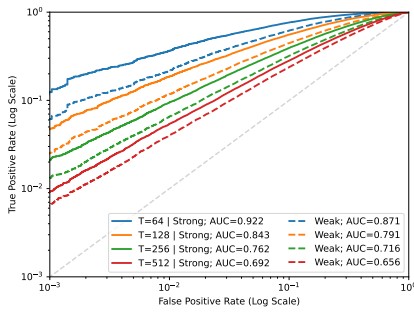
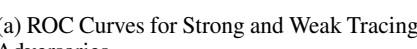

(a) ROC Curves for Strong and Weak Tracing Adversaries

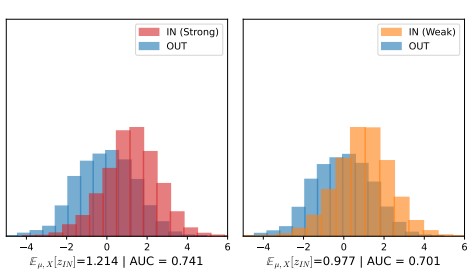

(b) Distribution of Test Statistic $z$

Figure 7: Results of Our Tracing Attack on Multitask Mean Estimation ( $T = 256$; $d = 256$; $N = 8$; $k = 4$ )

One possible attack would be the following:

1. The adversary receives a challenge set, or batch of data, $B$, from the challenger where $|B| = k \leq N$, such that the samples come from a task that was used to compute $\hat{\mu}$, $\mu_{IN} = \mu_i$, or a task that was *not* used to compute $\hat{\mu}$ but comes from the same underlying task distribution, $\mu_{OUT} \sim \mathcal{N}(\bar{\mu}, \bar{\sigma}^2 \mathbb{I}_d)$

2. First, the adversary computes the sample mean of the batch

$$\mu_B = \frac{1}{k} \sum_{x \in B} x$$

3. Then, the adversary computes the test statistic

$$z = \langle \hat{\mu} - \bar{\mu} \,,\, \mu_B - \bar{\mu} \rangle$$

4. Lastly, the adversary performs a thresholded classification, returning $z > \gamma$ for some threshold $\gamma$

Our adversary adapts a test statistic originally intended for membership-inference Dwork et al. (2015) that measures the correlation between the released statistic and the samples available to the adversary. However, unlike the membership-inference setting, the adversary is attempting to detect traces of the *task*, or data's distribution, $\mu_i$, rather than any particular sample from one of the datasets $X_i$. In the following theorems, we find that the inclusion of a task, $\mu_i$, can be inferred by both the **strong** and **weak** adversaries, and the **strong** adversary gets a slight advantage from having access to samples in $X_i$.

**Theorem B.1** (Strong Adversary; Theorem 4.1 in Main Body). *Let $\tau$ be the index of the target task and suppose that the challenger sends the adversary a challenge set of $k$ samples $B$ such that, when the task is **IN**, the $k$ samples are drawn uniformly at random from $X_\tau$. Then the expected value of the statistic, $z$, when $\mu_\tau$ is **OUT** is*

$$\mathbb{E}_{\mu, X} [z_{OUT}] = 0$$

*and when $\mu_\tau$ is **IN**, the expected value of $z$ is*

$$\mathbb{E}_{\mu, X} [z_{IN}] = \frac{d}{T} \left( \bar{\sigma}^2 + \frac{\sigma^2}{N} \right)$$

In Theorem 4.1, we see that the expectation of the **strong** adversary's test statistic, $z$, primarily grows in the dimension of the data divided by the total number of tasks used to learn the multitask mean, $\hat{\mu}$. Through this result, we demonstrate that detecting traces of tasks can be seen as a generalization of membership inference, as setting the number of tasks $T = 1$, sampling $N$ datapoints, $X$, only from the single task to estimate the task mean, and allowing the adversary to use $k = 1$ sample reduces $z_{IN}$ to the membership-inference test statistic, which has expectation $\Theta\left(\frac{d}{N}\right)$. In our setting, if the adversary wants to trace a single sample in the multitask mean, the expectation of their statistic

would be $\Theta\left(\frac{d}{NT}\right)$. Because the total dataset size for our mean estimation is $T \cdot N$ with $N \ll T$, tracing tasks is an easier objective for the adversary than tracing individual samples, as there a larger separation between the distributions of $z_{IN}$ and $z_{OUT}$. Moreover, because the dataset is composed of multiple task distributions, a **weak** task-inference adversary can mount attacks by using data that comes from one of the included tasks but was never used in the estimation of $\hat{\mu}$.

**Theorem B.2** (Weak Adversary; Theorem 4.2 in Main Body). *Let $\tau$ be the index of the target task and suppose the challenger sends the adversary a challenge set of $k$ samples $B$ such that, when the task is **IN**, the $k$ samples are drawn i.i.d. from the same distribution as $X_\tau$, $\mathcal{N}(\mu_\tau, \sigma^2 \mathbb{I}_d)$. Then the expected value of the statistic, $z$, when $\mu_\tau$ is **OUT** is 0. When $\mu_\tau$ is **IN**, the expected value of $z$ is*

$$\mathbb{E}_{\mu,X}[z_{IN}] = \frac{d}{T}\bar{\sigma}^2$$

Informally, Theorem 4.2 shows that the expectation of the weak adversary's test statistic depends only on the number of tasks, and does not benefit from a smaller number of total training samples. Similar to the **strong** case, when analyzing the **weak** case, we split the statistically dependent and independent components of the sum, as the batch is drawn from the underlying task distribution for some fixed task which was included in the estimate. In contrast with the **strong** case, there is no overlap between the adversary's challenge set and the training data. Thus, we need not account for overlapping $X_{\tau,j}$'s. Rather, we only need to consider that the challenge set samples come from the same distribution, or task, as one of the tasks in the dataset used to compute $\hat{\mu}$.

**Theorem B.3** (Variance of $z$; Theorem 4.3 in Main Body). *The variance of $z$ when $\mu_\tau$ is **OUT** is*

$$\text{Var}_{\mu,X}(z_{OUT}) = \frac{d}{T}\left[\bar{\sigma}^4 + \frac{\sigma^4}{kN} + \left(\frac{k+N}{kN}\right)(\bar{\sigma}^2\sigma^2)\right]$$

*When $\mu_\tau$ is **IN**,*

$$\text{Var}_{\mu,X}(z_{IN}) \leq 3\text{Var}_{\mu,X}(z_{OUT})$$

We note that the **strong** adversary always achieves greater attack success rates than the **weak** adversary because the distance between the expected value of the test statistic, $z$, when the task is **IN** or **OUT** is strictly larger than that of the **weak** adversary at a similar level of variance. Furthermore, this attack on mean estimation shows that the adversary's success is dependent on two distinct parts: the knowledge they receive from having *member* data (i.e. the $\frac{d}{TN}\sigma^2$ term) and the knowledge they have about the *distributions*, or *tasks*, that compose the dataset (i.e. the $\frac{d}{T}\bar{\sigma}^2$ term). To empirically verify our results, we simulated the attack with parameters $T = 256$, $d = 256$, $N = 8$, and $k = 4$. The ROC curves and distribution of $z$ for both the **strong** and **weak** adversaries are shown in Figure 7.

## B.2 Proofs for Section 4

In this section, we provide the proofs for our theorems in Section 4. Because the random variables in our estimation and attack are nested, we use the subscript $X$ to denote taking probability over sampling the data and subscript $\mu$ to denote taking probability over sampling tasks.

**Theorem 4.1** (Strong Adversary). *Let $\tau$ be the index of the target task and suppose that the challenger sends the adversary a challenge set of $k$ samples $B$ such that, when the task is **IN**, the $k$ samples are drawn uniformly at random from $X_\tau$. Then the expected value of the statistic, $z$, when $\mu_\tau$ is **OUT** is*

$$\mathbb{E}_{\mu,X}[z_{OUT}] = 0$$

and when $\mu_\tau$ is **IN**, the expected value of $z$ is

$$\mathbb{E}_{\mu,X}[z_{IN}] = \frac{d}{T}\left(\bar{\sigma}^2 + \frac{\sigma^2}{N}\right)$$

*Proof.* In the **strong** case, for tasks that were included, the adversary has access to samples which were used to compute the mean, $\hat{\mu}$. Suppose that the strong adversary computes $z$, then, when $\mu_\tau$ is **OUT**,

$$
\begin{aligned}
\mathop{\mathbb{E}}_{\mu,X} \left[ z_{OUT} \right] &= \mathop{\mathbb{E}}_{\mu,X} \left[ \langle \hat{\mu} - \bar{\mu} \, , \, \mu_B - \bar{\mu} \rangle \right] \\
&= \langle \mathop{\mathbb{E}}_{\mu,X} \left[ \hat{\mu} - \bar{\mu} \right] \, , \, \mathop{\mathbb{E}}_{\mu,X} \left[ \mu_B - \bar{\mu} \right] \rangle \\
&= 0
\end{aligned}
$$

In contrast, when the batch of points is **IN**,

$$
\begin{aligned}
\mathop{\mathbb{E}}_{\mu,X} \left[ z_{IN} \right] &= \mathop{\mathbb{E}}_{\mu,X} \left[ \langle \hat{\mu} - \bar{\mu} \, , \, \mu_B - \bar{\mu} \rangle \right] \\
&= \sum_{i=1}^{d} \mathop{\mathbb{E}}_{\mu,X} \left[ (\hat{\mu} - \bar{\mu})_i \cdot (\mu_B - \bar{\mu})_i \right]
\end{aligned}
$$

For succinctness, we drop the summation over $d$ dimensions as they are i.i.d.

$$
\begin{aligned}
\mathop{\mathbb{E}}_{\mu,X} \left[ (\hat{\mu} - \bar{\mu}) \cdot (\mu_B - \bar{\mu}) \right] &= \mathop{\mathbb{E}}_{\mu,X} \left[ \hat{\mu}\mu_B - \bar{\mu}\mu_B - \bar{\mu}\hat{\mu} + \bar{\mu}^2 \right] \\
&= \mathop{\mathbb{E}}_{\mu,X} \left[ \hat{\mu}\mu_B \right] - \bar{\mu} \mathop{\mathbb{E}}_{\mu,X} \left[ \mu_B \right] - \bar{\mu} \mathop{\mathbb{E}}_{\mu,X} \left[ \hat{\mu} \right] + \bar{\mu}^2 \\
&= \mathop{\mathbb{E}}_{\mu,X} \left[ \hat{\mu}\mu_B \right] - \bar{\mu}^2
\end{aligned}
$$

Now, expanding $\hat{\mu}$ and $\mu_B$

$$
\mathop{\mathbb{E}}_{\mu,X} \left[ \hat{\mu}\mu_B \right] - \bar{\mu}^2 = \mathop{\mathbb{E}}_{\mu,X} \left[ \left( \frac{1}{T} \sum_{i=1}^{T} \frac{1}{N} \sum_{j=1}^{N} X_{i,j} \right) \left( \frac{1}{k} \sum_{j=1}^{k} X_{\tau,j} \right) \right] - \bar{\mu}^2
$$

We assume that $k \leq N$. Separating the correlated and uncorrelated tasks, we have

$$
= \mathop{\mathbb{E}}_{\mu,X} \left[ \left( \frac{1}{T} \sum_{i \neq \tau} \frac{1}{N} \sum_{j=1}^{N} X_{i,j} \right) \left( \frac{1}{k} \sum_{j=1}^{k} X_{\tau,j} \right) \right]
$$

$$
+ \mathop{\mathbb{E}}_{\mu,X} \left[ \left( \frac{1}{TN} \sum_{j=1}^{N} X_{\tau,j} \right) \left( \frac{1}{k} \sum_{j=1}^{k} X_{\tau,j} \right) \right] - \bar{\mu}^2
$$

$$
= \mathop{\mathbb{E}}_{\mu,X} \left[ \left( \frac{1}{T} \sum_{i \neq \tau} \frac{1}{N} \sum_{j=1}^{N} X_{i,j} \right) \right] \cdot \mathop{\mathbb{E}}_{\mu,X} \left[ \left( \frac{1}{k} \sum_{j=1}^{k} X_{\tau,j} \right) \right]
$$

$$
+ \mathop{\mathbb{E}}_{\mu,X} \left[ \left( \frac{1}{TN} \sum_{j=1}^{N} X_{\tau,j} \right) \left( \frac{1}{k} \sum_{j=1}^{k} X_{\tau,j} \right) \right] - \bar{\mu}^2
$$

$$
= \frac{T-1}{T} \bar{\mu}^2 + \mathop{\mathbb{E}}_{\mu,X} \left[ \left( \frac{1}{TN} \sum_{j=1}^{N} X_{\tau,j} \right) \left( \frac{1}{k} \sum_{j=1}^{k} X_{\tau,j} \right) \right] - \bar{\mu}^2
$$

$$
= \frac{1}{TNk} \left( \sum_{j=1}^{k} \mathop{\mathbb{E}}_{\mu,X} \left[ X_{\tau,j}^2 \right] + \sum_{j \neq \ell} \mathop{\mathbb{E}}_{\mu,X} \left[ X_{\tau,j} \cdot X_{\tau,\ell} \right] - Nk\bar{\mu}^2 \right)
$$

Now, taking the expectation over sampling the data then taking the expectation over sampling tasks,

$$
= \frac{1}{TNk} \left( \sum_{j=1}^{k} \mathop{\mathbb{E}}_{\mu} \left[ \mu_\tau^2 + \sigma^2 \right] + \sum_{j \neq \ell} \mathop{\mathbb{E}}_{\mu} \left[ \mu_\tau^2 \right] - Nk\bar{\mu}^2 \right)
$$

$$
= \frac{1}{TNk} \left( k(\bar{\mu}^2 + \bar{\sigma}^2 + \sigma^2) + (Nk - k)(\bar{\mu}^2 + \bar{\sigma}^2) - Nk\bar{\mu}^2 \right)
$$

$$
= \frac{1}{TNk} \left( k\bar{\sigma}^2 + k\sigma^2 + (Nk - k)\bar{\sigma}^2 \right)
$$

$$
= \frac{1}{TNk} \left( Nk\bar{\sigma}^2 + k\sigma^2 \right)
$$

$$
= \frac{\bar{\sigma}^2}{T} + \frac{\sigma^2}{NT}
$$

Summing over the $d$ i.i.d. dimensions, we get

$$
\mathop{\mathbb{E}}_{\mu,X} [z_{IN}] = \frac{d}{T} (\bar{\sigma}^2 + \frac{\sigma^2}{N})
$$

$\square$

**Theorem 4.2** (Weak Adversary). Let $\tau$ be the index of the target task and suppose the challenger sends the adversary a challenge set of $k$ samples $B$ such that, when the task is **IN**, the $k$ samples are drawn i.i.d. from the same distribution as $X_\tau$, $\mathcal{N}(\mu_\tau, \sigma^2 \mathbb{I}_d)$. Then the expected value of the statistic, $z$, when $\mu_\tau$ is **OUT** is 0. When $\mu_\tau$ is **IN**, the expected value of $z$ is

$$\mathbb{E}_{\mu,X}[z_{IN}] = \frac{d}{T}\bar{\sigma}^2$$

*Proof.* The proof for the **OUT** case is identical to the proof for Theorem 4.1. When the **weak** adversary's challenge batch is **IN**

$$\mathbb{E}_{\mu,X}[z_{IN}] = \mathbb{E}_{\mu,X}[\langle \hat{\mu} - \bar{\mu} \, , \, \mu_B - \bar{\mu}\rangle]$$

$$= \sum_{i=1}^{d} \mathbb{E}_{\mu,X}[(\hat{\mu} - \bar{\mu})_i \cdot (\mu_B - \bar{\mu})_i]$$

For succinctness, we drop the summation over $d$ dimensions as they are i.i.d.

$$\mathbb{E}_{\mu,X}[(\hat{\mu} - \bar{\mu}) \cdot (\mu_B - \bar{\mu})] = \mathbb{E}_{\mu,X}\left[\hat{\mu}\mu_B - \bar{\mu}\mu_B - \bar{\mu}\hat{\mu} + \bar{\mu}^2\right]$$

$$= \mathbb{E}_{\mu,X}[\hat{\mu}\mu_B] - \bar{\mu}\mathbb{E}_{\mu,X}[\mu_B] - \bar{\mu}\mathbb{E}_{\mu,X}[\hat{\mu}] + \bar{\mu}^2$$

$$= \mathbb{E}_{\mu,X}[\hat{\mu}\mu_B] - \bar{\mu}^2$$

Expanding $\hat{\mu}$ and $\mu_B$ yields

$$\mathbb{E}_{\mu,X}[\hat{\mu}\mu_B] - \bar{\mu}^2 = \mathbb{E}_{\mu,X}\left[\left(\frac{1}{T}\sum_{i=1}^{T}\frac{1}{N}\sum_{j=1}^{N}X_{i,j}\right)\left(\frac{1}{k}\sum_{j=1}^{k}X_{\tau,j}\right)\right] - \bar{\mu}^2$$

We assume that $k \leq N$. Once again separating the correlated and uncorrelated tasks, we have

$$= \mathbb{E}_{\mu,X}\left[\left(\frac{1}{T}\sum_{i\neq\tau}\frac{1}{N}\sum_{j=1}^{N}X_{i,j}\right)\right] \cdot \mathbb{E}_{\mu,X}\left[\left(\frac{1}{k}\sum_{j=1}^{k}X_{\tau,j}\right)\right]$$

$$+ \mathbb{E}_{\mu,X}\left[\left(\frac{1}{TN}\sum_{j=1}^{N}X_{\tau,j}\right)\left(\frac{1}{k}\sum_{j=1}^{k}X_{\tau,j}\right)\right] - \bar{\mu}^2$$

$$= \frac{T-1}{T}\bar{\mu}^2 + \mathbb{E}_{\mu,X}\left[\left(\frac{1}{TN}\sum_{j=1}^{N}X_{\tau,j}\right)\left(\frac{1}{k}\sum_{j=1}^{k}X_{\tau,j}\right)\right] - \bar{\mu}^2$$

Now, we note that the challenge batch samples are fresh, i.i.d samples from the task distribution $\mathcal{N}(\mu_\tau, \sigma^2 \mathbb{I}_d)$. Thus, the challenge batch and $\hat{\mu}$ are independent over sampling the data, but correlated over tasks. Taking the expectation over both data and tasks,

$$= \frac{T-1}{T}\bar{\mu}^2 + \mathop{\mathbb{E}}_{\mu}\left[\frac{1}{T}\mathop{\mathbb{E}}_{X}\left[\frac{1}{N}\sum_{j=1}^{N} X_{\tau,j}\right] \cdot \mathop{\mathbb{E}}_{X}\left[\frac{1}{k}\sum_{j=1}^{k} X_{\tau,j}\right]\right] - \bar{\mu}^2$$

$$= \frac{T-1}{T}\bar{\mu}^2 + \frac{1}{T}\mathop{\mathbb{E}}_{\mu}\left[\mu_\tau^2\right] - \bar{\mu}^2$$

$$= \frac{T-1}{T}\bar{\mu}^2 + \frac{1}{T}(\bar{\mu}^2 + \bar{\sigma}^2) - \bar{\mu}^2$$

$$= \frac{1}{T}(\bar{\mu}^2 + \bar{\sigma}^2) - \frac{1}{T}\bar{\mu}^2$$

$$= \frac{\bar{\sigma}^2}{T}$$

Summing over the $d$ i.i.d. dimensions, we get that under the **weak** adversary

$$\mathop{\mathbb{E}}_{\mu,X}[z_{IN}] = \frac{d}{T}\bar{\sigma}^2$$

$\square$

**Theorem 4.3 (Variance).** The variance of $z$ when $\mu_\tau$ is **OUT** is

$$\mathop{\mathrm{Var}}_{\mu,X}(z_{OUT}) = \frac{d}{T}\left[\bar{\sigma}^4 + \frac{\sigma^4}{kN} + \left(\frac{k+N}{kN}\right)(\bar{\sigma}^2\sigma^2)\right]$$

When $\mu_\tau$ is **IN**,

$$\mathop{\mathrm{Var}}_{\mu,X}(z_{IN}) \leq 3\mathop{\mathrm{Var}}_{\mu,X}(z_{OUT})$$

*Proof.* First, we define the following random variables:

$$\alpha = \hat{\mu} - \bar{\mu}; \; \alpha \sim \mathcal{N}\left(\vec{0}, \; \left(\frac{\bar{\sigma}^2}{T} + \frac{\sigma^2}{NT}\right)\mathbb{I}_d\right)$$

$$\beta = \mu_B - \bar{\mu}; \; \beta \sim \mathcal{N}\left(\vec{0}, \; \left(\bar{\sigma}^2 + \frac{\sigma^2}{k}\right)\mathbb{I}_d\right)$$

Then, the variance in the **OUT** case is

$$\mathop{\mathrm{Var}}_{\mu,X}(z_{OUT}) = \mathop{\mathrm{Var}}_{\mu,X}(\langle \alpha \, , \, \beta \rangle)$$

$$= \sum_{i=1}^{d} \mathop{\mathrm{Var}}_{\mu,X}(\alpha_i \cdot \beta_i)$$

For succinctness, we drop the summation with index $i$ over the i.i.d dimensions of the random variables. Then, we have

$$
\begin{aligned}
\operatorname*{Var}_{\mu,X}(\alpha \cdot \beta) &= \operatorname*{\mathbb{E}}_{\mu,X}\left[\alpha^2\beta^2\right] - \operatorname*{\mathbb{E}}_{\mu,X}[\alpha\beta]^2 \\
&= \operatorname*{\mathbb{E}}_{\mu,X}\left[\alpha^2\beta^2\right] \\
&= \operatorname*{\mathbb{E}}_{\mu,X}\left[\alpha^2\right]\operatorname*{\mathbb{E}}_{\mu,X}[\beta2] \\
&= \operatorname*{Var}_{\mu,X}(\alpha)\operatorname*{Var}_{\mu,X}(\beta) \\
&= \left(\frac{\bar{\sigma}^2}{T} + \frac{\sigma^2}{NT}\right) \cdot \left(\bar{\sigma}^2 + \frac{\sigma^2}{k}\right) \\
&= \frac{1}{T}\left[\bar{\sigma}^4 + \frac{\sigma^4}{kN} + \left(\frac{k+N}{kN}\right)(\bar{\sigma}^2\sigma^2)\right]
\end{aligned}
$$

Summing over the $d$ dimensions, we have

$$
\operatorname*{Var}_{\mu,X}(z_{OUT}) = \frac{d}{T}\left[\bar{\sigma}^4 + \frac{\sigma^4}{kN} + \left(\frac{k+N}{kN}\right)(\bar{\sigma}^2\sigma^2)\right]
$$

Now, we can bound the variance of $z_{IN} = \langle \alpha , \beta \rangle$

$$
\begin{aligned}
\operatorname*{Var}_{\mu,X}(z_{IN}) &= \operatorname*{Var}_{\mu,X}(\langle \alpha , \beta \rangle) \\
&= \sum_{i=1}^{d}\operatorname*{Var}_{\mu,X}(\alpha_i \cdot \beta_i) \\
&= \sum_{i=1}^{d}\operatorname*{\mathbb{E}}_{\mu,X}\left[(\alpha_i \cdot \beta_i)^2\right] - \operatorname*{\mathbb{E}}_{\mu,X}[\alpha_i \cdot \beta_i]^2 \\
&= \sum_{i=1}^{d}\operatorname*{\mathbb{E}}_{\mu,X}\left[(\alpha_i \cdot \beta_i)^2\right] - \operatorname*{Cov}_{\mu,X}(\alpha_i, \beta_i)^2 \\
&= \sum_{i=1}^{d}\operatorname*{\mathbb{E}}_{\mu,X}\left[(\alpha_i \cdot \beta_i)^2\right] - \rho^2\operatorname*{Var}_{\mu,X}(\alpha_i)\operatorname*{Var}_{\mu,X}(\beta_i)
\end{aligned}
$$

where $\rho$ is the correlation coefficient between $\alpha_i$ and $\beta_i$. Now, using the Cauchy-Schwarz inequality and the fact that $\rho^2 \geq 0$

$$
\begin{aligned}
&\leq \sum_{i=1}^{d}\left(\sqrt{\operatorname*{\mathbb{E}}_{\mu,X}\left[\alpha_i^4\right]\operatorname*{\mathbb{E}}_{\mu,X}\left[\beta_i^4\right]}\right) - \rho^2\operatorname*{Var}_{\mu,X}(\alpha_i)\operatorname*{Var}_{\mu,X}(\beta_i) \\
&= \sum_{i=1}^{d}\left(\sqrt{3\operatorname*{Var}_{\mu,X}(\alpha_i)^2 \cdot 3\operatorname*{Var}_{\mu,X}(\beta_i)^2}\right) - \rho^2\operatorname*{Var}_{\mu,X}(\alpha_i)\operatorname*{Var}_{\mu,X}(\beta_i) \\
&= \sum_{i=1}^{d}3\operatorname*{Var}_{\mu,X}(\alpha_i)\operatorname*{Var}_{\mu,X}(\beta_i) - \rho^2\operatorname*{Var}_{\mu,X}(\alpha_i)\operatorname*{Var}_{\mu,X}(\beta_i) \\
&\leq \sum_{i=1}^{d}3\operatorname*{Var}_{\mu,X}(\alpha_i)\operatorname*{Var}_{\mu,X}(\beta_i) \\
&= 3\operatorname*{Var}_{\mu,X}(z_{OUT})
\end{aligned}
$$

We note that in this proof, the variance of $z_{IN}$ is bounded by setting the squared correlation equal to 0. In practice, $\rho^2 > 0$ since we know from Theorems 4.1 and 4.2 that $\mathbb{E}_{\mu,X}[\alpha_i \cdot \beta_i] = \frac{\bar{\sigma}^2}{T}$ in the **weak** case and $\frac{\bar{\sigma}^2}{T} + \frac{\sigma^2}{NT}$ in the **strong** case.

$\square$

## C ADDITIONAL DISCUSSION

We provide additional discussion points to supplement Section 7.

### C.1 POTENTIAL DEFENSES

Throughout our evaluation, we use gradient normalization and clipping, along with weight decay (or $L_2$ regularization) to improve the stability of MTL with few samples per task. In contrast to observations for empirical defenses made in the membership-inference literature Carlini et al. (2022), our task-inference attack succeeds even when the target MTL model is trained using gradient clipping. We find that these heuristic methods to mitigate overfitting are not sufficient to defend against task-inference.

Thus, we evaluate differential privacy (Dwork et al., 2006; Abadi et al., 2016) with *task-level guarantees* as a certified defense. This corresponds to user-level (or group-level) differential privacy, where neighboring datasets are defined by the inclusion or omission of a user's entire contribution to the dataset, rather than an individual sample. While there are several works studying user-level differential privacy when training machine learning models Levy et al. (2021); Chua et al. (2024); Charles et al. (2024) and estimating high dimensional means Cummings et al. (2022); Agarwal et al. (2025), only one work develops algorithms with *client-level* privacy guarantees Hu et al. (2023) when MTL is applied to collaborative (or federated) learning.

In this set of experiments, we instantiate a simple version of task-level DP training by clipping the aggregate task gradients, which bounds any given task's influence, before adding Gaussian noise to the average update of the *shared* parameters. The task heads are not trained with DP as they are only trained on their corresponding task's data. We focus on repeating the vision experiments on ResNet-8 and FEMNIST discussed in Section A.6, but now with formal privacy guarantees. We choose FEMNIST (and the small ResNet-8 model) in particular because the dataset provides a sufficient number of samples per task to maintain non-trivial model utility under the constraints of DP. Table 3 shows that while task-level DP mitigates task-inference risk, it incurs a prohibitive cost to model utility. In particular, even at a relatively high privacy budget of $\varepsilon = 16$, the small amount of added noise degrades the quality of the shared representation, causing the model's top-1 accuracy to drop to 29%. These results reinforce our findings in Section 6.5, which show that the task-inference success heavily depends on the utility of the learned representation.

Table 3: Task-Inference Success (AUC) and Model Utility (Top-1 Acc.) with Task-Level Differential Privacy on FEMNIST.

| | | | Variance Attack AUC | |
|---|---|---|---|---|
| Dataset | Privacy Budget ($\varepsilon$) | Top-1 Acc. | Strong | Weak |
| | No Noise ($\varepsilon = \infty$) | 71% | 0.69 | 0.57 |
| FEMNIST | $\varepsilon = 16.0$ | 29% | 0.49 | 0.48 |
| | $\varepsilon = 8.0$ | 26% | 0.49 | 0.49 |

### C.2 LACK OF SUITABLE BASELINES

Prior (sample-level) inference attacks on encoder models Liu et al. (2021b) exploit the tendency of well-trained encoders to produce similar embeddings for augmentations of the same training image, and rely on shadow models trained on labeled reference data to calibrate their attack. In contrast, our adversary has strictly weaker access: given an input sample, they receive only the corresponding

embedding vector output by the shared representation. Neither of the adversaries we consider has knowledge of the model's weights, task-specific classification heads, nor the ability to compute per-sample loss or confidence scores. This renders baselines, such as aggregated membership-inference, ill-suited for direct comparison since even purely black-box membership inference attacks rely on statistics derived from full classifier outputs. The task-inference adversary in our threat model queries the shared representation once per sample as a black-box function that maps inputs to embeddings. Our attacks leverage the implicitly learned dependencies and correlations between embedding vectors from the same task.

## D    USE OF LARGE LANGUAGE MODELS

Large language models were used to polish writing by rephrasing sentences, helping condense long paragraphs, and checking for typos and grammatical errors.

