# OpenReview forum: "Black-Box Privacy Attacks on Shared Representations in Multitask Learning"
_ICLR.cc/2026/Conference — ICLR 2026 Poster_

### Official Review · Reviewer_Yemp · 2025-10-27

[review text omitted: it was posted to a different submission]

---

> ### Author Response · Authors · 2025-11-25
> **Response to Reviewer Yemp**
>
> We thank Reviewer Yemp for their review and comments. We have attached the revised version of our paper (with revisions in blue) and address your comments and concerns below:
>
>
> 1. **Adversary’s Goals:**
>  The review states we demonstrate "qualitative image reconstructions but lack rigorous privacy leakage measurement." However, our paper does not focus on data reconstruction attacks; rather, we focus on inferring the membership of tasks in the training data.
> Regarding the reviewer's claim that the paper “lacks rigorous privacy leakage measurement” (such as membership-inference accuracy) we highlight that we exclusively report standard, quantitative metrics for measuring the success of inference attacks, such as ROC curves, AUC scores, and TPR at low FPR.
>
> 2. **Models and Evaluation:**
> The review claims we "train a small decoder network" using "auxiliary generative priors”. We do not train decoder networks or use generative priors. Our attacks are simple, efficient tests calculated directly on the shared embeddings. They do not require strong assumptions or training of reference models.
>
> 3. **Baselines:**
> The review suggests comparing our method to Deep Leakage from Gradients (DLG) [1]. DLG is a white-box reconstruction attack. Since our goal is black-box, task-level membership-inference, DLG is not a relevant baseline.
>
>
> 4. **Larger Models:**
> We show our attacks’ ability to scale to large architectures and the text generation domain. In addition to our BERT evaluation on Stack Overflow, we have added new natural language results for Gemma 3 270M fine-tuned using MTL on the Reddit dataset (Section 6.3) for personalized next-token prediction. We fully fine-tune the shared weights of Gemma 3 and learn task-specific adapters on the language modeling head. Figure 1d shows the ROC curves for these experiments. In summary, the strong adversary achieves a task-inference AUC of 0.84 and 0.73, while the weak adversary achieves a task-inference AUC of 0.77 and 0.66 using the variance and inner product attacks, respectively.
>
> [1] Zhu et al., *Deep Leakage from Gradients* (NeurIPS 2019)

---

> > ### Comment · Reviewer_Yemp · 2025-11-26
> >
> > Thanks for the response. This work explores task-level inference in MTL. While some white-box attacks are not applicable in the proposed setting, a brief discussion about the comparison of existing black-box attacks should also be included.
> >
> > Besides, although the "black box setting" is proposed, the adversary has access to some exact training samples of the target task, which is rarely realistic in realistic applications. Could you provide some discussions?

---

> ### Comment · Area_Chair_LPpV · 2025-11-26
>
> Regarding the ethical aspects, we note that the reviewer’s concerns relate to technical assumptions (comparison with prior black-box attacks and realism of the adversary having training samples) rather than ethical risks. The paper uses only public datasets and does not interact with identifiable personal data. Thus no additional ethics issues arise; the authors are encouraged to address the reviewer’s technical questions in the rebuttal.

---

> ### Author Response · Authors · 2025-11-27
> **Response to Reviewer Yemp**
>
> Thank you for your response.
>
> Regarding baselines, reconstruction is a fundamentally different and harder objective than membership inference. While perfect reconstruction of training samples implies membership-inference, there isn’t an established method for using imperfect reconstruction as a basis for membership-inference, and it wouldn’t provide an interesting baseline.  Furthermore, concrete reconstruction attacks require a *significantly* more powerful adversary and are ill-suited as a direct baseline for our task-inference adversary’s access and goals. We discuss how our proposed threat model compares and contrasts with existing work in Section 2 of the paper.
>
> The standard membership-inference threat model necessitates that the adversary has access to the exact training data of a “person” to correctly identify people who are in the dataset. The standard way to generalize this methodology to our setting, where one person is a task, is to give the attacker all or some of the exact training data for a task.  We also address this limitation and propose the "weak adversary” who uses the distributional structure of tasks to mount semantically meaningful membership-inference attacks on entire tasks *without* access to exact training samples. For example, if images of a given user’s face were used to train a shared representation, the weak adversary could mount a successful task-inference attack using *new, unseen* photos of the user’s face. We formalize the separation between the cases where the adversary does or does not have access to exact training examples using Gaussian mean estimation in Section 4, and we empirically validate this separation in every experiment throughout Section 6.
>
> We are happy to add a subsection to the final paper to explicitly differentiate our approach from existing data reconstruction attacks and white-box approaches.

---

### Official Review · Reviewer_cjJ7 · 2025-11-01

**Soundness:** 3
**Presentation:** 4
**Contribution:** 3
**Rating:** 6
**Confidence:** 2

**Summary:**

The paper investigates privacy leakage in the shared representations from Multitask learning. To do this, the paper presents a black box attach which tries to infer the inclusion of a task during MTL. This is done by assuming an adversary has access to samples from the target task's distribution (either from the training set, a strong adversary, or independently from the target task distribution; a week adversary). and queries the shared representation of the model. Experiments demonstrate the success of the attack across vision and language settings.

**Strengths:**

* The paper instroduces a novel task-inference threat model that generalizes the sample-level membership inference to the task level.

* The attachs presented a black box and is not limited by the requirement of shadow models nor labeled reference data.

* The empirical evaluations span both vision and language areas and use multiple datasets (CelebA, FEMNIST, StackOverflow) and MTL two common MTL cases (personalization and multi-problem learning).

* The attack algorithms are clearly described, and the results highlight that the non-trivial success of the attack.

**Weaknesses:**

* It is not clear how this novel attack compares to existing relevant baselines in these inference attachks (such as potentially those requiring stronger assumptions for the adversary).

**Questions:**

* How is the relationship between the attack simplicity and its attack power in the context of existing methods from prior work?

* Can the paper comment on how some existing defenses mitigate the task-inference risk?

---

> ### Author Response · Authors · 2025-11-25
> **Response to Reviewer cjJ7**
>
> We thank Reviewer cjJ7 for their review and appreciate their comments and feedback. We have attached the revised version of our paper (with revisions in blue) and address your comments and concerns below:
>
> 1. **Baselines:**
> Existing works do not study our proposed task-inference threat model (i.e. black-box access to the shared representation) and, thus, do not provide comparable baselines. For example, prior attacks on MTL [1] assume access to task-specific heads, which would make task-inference trivial.
> One possible baseline would be to aggregate sample-level membership-inference attacks [2] in order to determine an entire task’s inclusion in the dataset. However, such an attack would only use the signal from a single, random vector in the embedding space to compute the test statistic, which makes defending against these attacks trivial. For example, if an adversary was using the $\ell_1$ norm as a way of detecting “overconfident” activations, the model owner could simply transform the embeddings to have norm equal to 1 while maintaining the model’s utility. Additionally, the weak adversary, which only uses out-of-sample data, would fail using this baseline, as the baseline MIA is attempting to infer which samples actually appeared in the training set.
>
> 2. **Attack Simplicity and Distinguishing Power:**
> Our attack is purely black-box, taking less than a second per task-inference query. Unlike prior works that rely on training expensive shadow models to calibrate their decision thresholds, we leverage the group, or distributional, structure of data belonging to the same task. By measuring and aggregating the codependencies across multiple embeddings from the same task, we amplify the attack’s signal to determine a task’s membership without the need for labeled reference data.
>
>
> 3. **Defenses:**
> We have added Appendix C.1 to evaluate differential privacy (DP) with task-level guarantees as a defense against our task-inference attacks. As shown in Table 3, while DP mitigates the attack, it induces a large utility penalty (71% top-1 accuracy to 29% top-1 accuracy @ $\varepsilon = 16$), which renders the model unusable. This aligns with our finding in Section 6.5: the success of task-inference heavily depends on the utility of the shared representation. We note that all of the models in our experiments are trained with both gradient clipping and weight decay which highlights the insufficiency of heuristic defenses.
>
> [1] Yan et al., *MTL-Leak: Privacy Risk Assessment in Multi-Task Learning* (IEEE Transactions on Dependable and Secure Computing)
> [2] Maini et al., *Dataset Inference: Ownership Resolution in Machine Learning* (ICLR 2021)

---

### Official Review · Reviewer_2x6h · 2025-11-01

**Soundness:** 3
**Presentation:** 3
**Contribution:** 2
**Rating:** 6
**Confidence:** 4

**Summary:**

This paper introduces a novel privacy threat called task-inference in Multitask Learning (MTL). The authors investigate whether a shared representation, the minimal component required for joint training in MTL, leaks information about the specific tasks used to train it. They propose a purely black-box threat model where an adversary, with only query access to the shared representation, aims to determine if a given task was included in the training set. The paper develops two efficient attacks, a Coordinate-Wise Variance Attack and a Pairwise Inner Product Attack, which exploit the codependencies of embeddings from the same task without requiring shadow models or auxiliary reference data. Experiments across vision (CelebA, FEMNIST) and language (Stack Overflow) datasets demonstrate that these attacks are effective, even for a "weak" adversary who only has access to fresh samples from a task, not the original training data. The paper links this leakage to the model's generalization gap, suggesting that models memorize properties of entire task distributions.

**Strengths:**

- This paper presents a novel and highly relevant threat model, "task-inference," which generalizes sample-level membership inference to the task level. This is a more practical and realistic threat for collaborative learning paradigms like MTL and federated learning, where the unit of privacy is often an entire user or data silo.
- The proposed attacks are efficient and operate under minimal adversarial assumptions. A significant strength is that they are purely black-box and do not require training costly shadow models or having access to large, labeled auxiliary datasets, which are common barriers for other inference attacks.
- The paper provides strong theoretical motivation for the attacks by analyzing an analogous tracing attack in a simplified Gaussian mean estimation problem. This analysis formally explains the observed performance gap between "strong" and "weak" adversaries.
- The experimental evaluation is comprehensive, validating the attacks across both vision and language domains (ResNet on CelebA/FEMNIST and BERT on Stack Overflow).
- The authors test the attacks in two distinct and practical MTL scenarios: personalization (where a task is a user) and solving multiple learning problems (where a task is a distinct classification objective), demonstrating the broad applicability of the threat.

**Weaknesses:**

- The paper does not clearly state whether the primary experimental results (e.g., in Figure 1 and 2) are averaged over multiple independent MTL training runs. While ablations mention multiple runs, the stability of the main attack results against different model initializations is not explicitly confirmed.
- The practical utility of the "blind" percentile-based thresholds is questionable. As shown in Table 1, these thresholds can result in very high false positive rates (e.g., 47.6% or 41.7%), which would make the attack unreliable for an adversary who truly has no reference data to set a better threshold.
- The "Pairwise Inner Product Attack" uses a whitening transformation that is computed by pooling all available embeddings. This seems to imply the adversary needs access to a broad set of embeddings from many different tasks, which somewhat weakens the "minimal knowledge" claim. The paper does not ablate the impact of this step.
- The paper’s discussion of defenses is very limited. It mentions user-level differential privacy (DP) but does not evaluate it or any other potential mitigation, such as representation compression or other regularization techniques beyond the gradient clipping that was already used during training.
- There is no comparison against adapted baselines. For instance, it's unclear how these novel attacks would perform against a simpler approach, like running a standard sample-level membership inference attack on all of the adversary's samples and aggregating the results (e.g., via majority vote) to get a task-level prediction.

**Questions:**

- Could the authors clarify how many independent MTL training runs were used for the main experiments in Figures 1 and 2? Are the reported ROC curves and AUC scores averaged across these runs?
- How critical is the whitening transformation to the "Pairwise Inner Product Attack"? How many tasks must an adversary pool embed to compute a stable covariance matrix, and how does the attack perform without this step?
- Have the authors considered comparing their attacks to a baseline where a standard, sample-level membership inference attack's predictions are aggregated (e.g., by averaging confidence scores or a majority vote) to produce a task-level inference?
- For the "multiple learning problems" case on Stack Overflow, the performance gap between strong and weak adversaries nearly disappears. Does this imply that any high-utility MTL model where tasks are defined by distinct labels will be inherently vulnerable to task inference, even from adversaries with no access to training data?
- L2​ decay or dropout on the shared representation be effective mitigations, or are formal defenses like user-level DP essential?

---

> ### Author Response · Authors · 2025-11-25
> **Response to Reviewer 2x6h**
>
> We thank Reviewer 2x6h for their review and appreciate both their comments on the relevance of our proposed threat model and feedback regarding our evaluation. We have attached the revised version of our paper (with revisions in blue) and address your comments and concerns below:
>
>
> 1. **Evaluation Details:**
> We aggregate results (shown in Figures and Tables 1 & 2) over 4 MTL training runs and 8 MTL training runs for our experiments on personalization and multiple learning problems, respectively. After each training run, we subsample the adversary’s task data and run 128 - 256 trials of the attack.
>
>
> 2. **Whitening:**
> While the whitening step in our attack *does* require some amount of auxiliary data, the adversary has no knowledge of whether the auxiliary data belongs to IN or OUT tasks. The adversary can use any publicly available, relevant images or text samples from the web to estimate the “global covariance” of the embedding space. Since the embedding space is typically low-dimensional, estimating the covariance matrix does not require too many samples (# of samples $\approx$ the embedding dimension squared for high accuracy).
>
> 3. **MIA as a Baseline:**
> While an aggregated membership inference attack could indeed succeed at task-inference (see [1]), these attacks do not incorporate the task’s structure in their test statistic. Such an adversary would only use the signal from a single, random vector in representation space to compute their test statistic, which makes defending against these attacks trivial. For example, if an adversary was using the $\ell_1$ norm as a way of detecting “overconfident” activations, the model owner could simply transform the embeddings to have norm equal to 1 while maintaining the model’s utility. Additionally, the weak adversary, which only uses out-of-sample data, would fail using this baseline, as the baseline MIA is attempting to infer which samples actually appeared in the training set.
>
>
> 4. **Performance Gap on Stack Overflow:**
> In the "Multiple Learning Problems" setting, the tasks are explicitly defined by the labeling of the data (e.g., the topic "Python"). In order for MTL to maximize utility, the shared model must learn the features associated with correctly classifying any given label in the training dataset. Thus, successful classification directly implies successful task-inference, which results in a much smaller gap between the strong and weak adversaries.
>
> 5. **Exploration of Defenses:**
> We note that all of the models in our experiments are trained with both gradient clipping and weight decay which highlights the insufficiency of heuristic defenses. Thus, we have added Appendix C.1 to evaluate differential privacy (DP) with task-level guarantees as a defense against our task-inference attacks. As shown in Table 3, while DP mitigates the attack, it induces a large utility penalty (71\% top-1 accuracy to 29% top-1 accuracy @ $\varepsilon = 16$), which renders the model unusable. This aligns with our finding in Section 6.5: the success of task-inference heavily depends on the utility of the shared representation.
>
> 6. **Blind Thresholds**
> Table 1 shows that, while increasing the threshold to the 90th percentile decreases average-case balanced accuracy (e.g., from 80% to 59.5% on CelebA for the strong adversary), the higher threshold offers more desirable trade-offs between TPR and FPR. For example, the 50th percentile threshold yields a TPR $\approx 4 \times$ FPR, while the 90th percentile threshold gives a TPR $\approx 39 \times$ FPR. This reinforces our claim that a task-inference adversary can minimize false positives with minimal knowledge and without calibrating their decision threshold using shadow models.
>
> [1] Maini et al., *Dataset Inference: Ownership Resolution in Machine Learning* (ICLR 2021)

---

### Official Review · Reviewer_LGXe · 2025-11-03

**Soundness:** 3
**Presentation:** 3
**Contribution:** 3
**Rating:** 4
**Confidence:** 4

**Summary:**

The paper introduces a new privacy attack in the multi-task learning (MTL) framework, where a shared representation is learned from data across multiple tasks. The goal of the attack is to infer whether a given task was included in the model’s training phase. Unlike traditional membership inference attacks at the sample level, this work operates at the task level, making it a higher-level privacy threat. The proposed attack is black-box and notably requires no auxiliary data or shadow models.
Two adversary settings are considered:
Strong adversary, which has access to some task data or samples drawn from the task’s distribution.
Weak adversary, which lacks access to such samples.

**Strengths:**

- The paper introduces a new threat model that examines how shared representations in multi-task learning can lead to privacy leakage.
- It presents a black-box attack formulated under both weak and strong adversary assumptions, which makes the study applicable to real-world conditions.
- A theoretical discussion is provided to explain the connection between representation sharing and privacy risks, highlighting the main factors that affect the effectiveness of the attack.

**Weaknesses:**

- There is a noticeable difference in performance between the weak and strong adversary scenarios.
- The paper does not explore or evaluate any potential defense strategies against the proposed attack.
- The method used to determine the attack’s decision threshold seems ad hoc, with limited justification or analysis of how it might perform in more realistic environments.

**Questions:**

- Can applying DP-SGD or other differential privacy methods mitigate the attack’s success rate?
- Is the proposed attack scalable to larger and more complex models such as transformers or LLM-scale encoders?

---

> ### Author Response · Authors · 2025-11-25
> **Response to Reviewer LGXe**
>
> We thank reviewer LGXe for their comments and suggestions regarding potential defenses and scalability of our attack. We have attached the revised version of our paper (with revisions in blue) and address your concerns below:
>
> 1. **Strong/Weak Adversary Performance Gap:**
> This gap is expected and consistent with our theoretical analysis in Section 4. The strong adversary benefits from both membership-inference signal *and* distribution-inference signal, while the weak adversary relies entirely on distribution-level memorization. In spite of this gap, the weak adversary succeeds across all of our experiments while only having access to samples which were not included in the training dataset of the MTL model.
>
>
> 2. **DP-SGD as a Defense:**
> We have added Appendix C.1 to evaluate differential privacy (DP) with task-level guarantees as a defense against our task-inference attacks. As shown in Table 3, while DP mitigates the attack, it induces a large utility penalty (71% top-1 accuracy -> 29% top-1 accuracy @ epsilon = 16), which renders the model unusable. This aligns with our finding in Section 6.5: the success of task-inference heavily depends on the utility of the shared representation. We note that all of the models in our experiments are trained with both gradient clipping and weight decay which highlights the insufficiency of heuristic defenses.
>
>
> 3. **Choosing Decision Thresholds:**
> Throughout the paper, we sweep over all possible thresholds to generate ROC curves which help us understand the achievable TPR/FPR trade-offs of our attacks. We explicitly use percentile thresholds (shown in Tables 1 and 2) to demonstrate how an adversary would perform in a strict black-box setting *without* the ability to calibrate their attack on known IN/OUT data. In this setting, the adversary receives a 1-dimensional statistic per task and can only draw an arbitrary, linear decision threshold. The ability of our task-inference adversary to distinguish IN and OUT tasks using these simple thresholds highlights the robustness of the attacks.
>
>
> 4. **Scalability to Transformers and LLMs:**
> We show our attacks’ ability to scale to large architectures and the text generation domain. In addition to our BERT evaluation on Stack Overflow, we have added new natural language results for Gemma 3 270M fine-tuned using MTL on the Reddit dataset (Section 6.3) for personalized next-token prediction. We fully fine-tune the shared weights of Gemma 3 and learn task-specific adapters on the language modeling head. Figure 1d shows the ROC curves for these experiments. In summary, the strong adversary achieves a task-inference AUC of 0.84 and 0.73, while the weak adversary achieves a task-inference AUC of 0.77 and 0.66 using the variance and inner product attacks, respectively.

---

### Author Response · Authors · 2025-12-03
**Summary for Area Chairs**

Dear Area Chair,

Given the reassignment of ACs, we wanted to provide a summary of the reviewers' concerns, along with our responses and revisions we made to the paper to address them.

**Scalability of Our Attack (Reviewers LGXe, Yemp):** Reviewers asked whether our attack could scale from classification models to larger architectures like LLMs. Thus, we added Section 6.3 to evaluate our attack on text generation models. In our evaluation, we fine-tuned Gemma 3 270M using MTL on Reddit posts from 256 users for personalized next-token prediction. The results (Figure 1d) demonstrate that both the strong and weak adversaries can achieve high task-inference AUC (0.84 and 0.77, respectively) in this domain.

**Defenses (Reviewers LGXe, 2x6h, cjJ7):** Reviewers requested discussion of potential defenses. We added Appendix C.1 to evaluate differential privacy with task-level guarantees as a defense to our task-inference attack. The results show that while training with differential privacy successfully mitigates the attack, it destroys model utility (71% classification accuracy drops to 29% for $\varepsilon=16$). This aligns with our finding in Section 6.5: the success of task-inference heavily depends on the utility of the shared representation. Additionally, we highlight the insufficiency of heuristic defenses by noting that all of the models in our evaluation are trained with weight decay, data augmentation, and gradient clipping.


**Comparison to Baselines (Reviewers 2x6h, cjJ7, Yemp):** Reviewers asked about comparisons to existing baselines and aggregated sample-level membership inference. We clarified the following:

1.  Existing works do not study our proposed task-inference threat model (i.e. black-box access to the shared representation) and, thus, do not provide comparable baselines. For example, prior attacks on MTL assume access to task-specific heads, which would make task-inference trivial.

2.  Because our threat model assumes pure black-box access to the shared representation, a sample-level membership-inference adversary would rely on a single, random vector to compute their test statistic. This limitation makes defending against such attacks trivial.

3.  Baselines based on sample-level membership-inference would fail for the "weak adversary" because this task-inference threat model only requires fresh, out-of-sample, task data while membership-inference needs exact training points.




**Strong vs Weak Adversary Performance Gap (Reviewer LGXe):** The reviewer noted the performance gap between the strong and weak adversary settings. We clarified that this is consistent with our theoretical analysis in Section 4, where the strong adversary benefits from both membership and distribution signals. Despite the gap, the weak adversary successfully infers task inclusion using only out-of-sample data.

**Choosing Decision Thresholds (Reviewers 2x6h, LGXe):** Reviewers raised concerns about the way in which we choose the decision threshold for our attack’s test statistic. We provided the following clarifications:

1.  Throughout the evaluation, we sweep over all possible thresholds to generate ROC curves which show the achievable TPR/FPR trade-offs.

2.  Percentile-based thresholds used in Tables 1 and 2 demonstrate how an adversary would perform in a strict black-box setting without the ability to calibrate their attack on labeled reference data.

**Whitening (Reviewer 2x6h):** The reviewer asked whether applying a whitening transformation to embeddings was necessary for our attack and how this impacts our claim that the adversary has "minimal knowledge". We clarified that the adversary can estimate the "global covariance" of the embedding space using a relatively small amount of publicly available, unlabeled reference data (e.g., ImageNet or Wikitext). The whitening transformation is not strictly necessary, but it improves the adversary's distinguishing power.

---

### Meta-Review · Area_Chair_3mYR · 2025-12-16

**Summary:**

The reviewers generally commend the paper on introducing a relevant carefully formulated novel threat model and clearly formulated attacks.

Noted weaknesses include:
1. Performance difference between weak and strong adversaries.
2. Lack of evaluation of defence strategies.
3. Limited discussion on choosing attack threshold and concerns on performance on blindly chosen thresholds.
4. Scalability of the attack to larger models.
5. Lack of discussion of experimental repeats.
6. Concerns on use of whitening.
7. Lack of comparison against adapted baselines, e.g. aggregated sample-level MIA.

The review of Reviewer Yemp lists a number of further weaknesses, but I find it hard to connect these with the paper contents, which makes me wonder whether the reviewer has accidentally posted the review for a wrong paper.

**Reviewer Concerns:**

The authors have provided brief responses to all concerns raised by the reviewers along with a revised manuscript that incorporates many of the comments.

I feel that the responses generally address the concerns raised by the reviewers, but feel that the following points could be further improved:

#2. The evaluation of DP training uses privacy levels that significantly reduce the model accuracy while completely stopping the attack (attack performance less than random guessing). It would be interesting to see whether there would be a higher $\varepsilon$ setting that would still stop the attack while maintaining higher accuracy on the main task.

#4. The Gemma 3 270M model used to demonstrate scalability is not large by modern standards, but this seems hardly critical and I appreciate that the authors seem to be working with limited compute.

#7. I do not find the argument of not using standard MIA as a baseline entirely satisfying. The reviewers suggested aggregating MIA scores over the entire task, so claiming that this would not work because the attacker only has a single target vector feels like attacking a straw man. I agree that applying MIA is not trivial because the adversary only sees the shared representation mapping and not a full classifier, but the shared representation is still a function rather than a single vector like the authors' response would indicate.

**Reviewer Scores:**

I am excluding Reviewer Yemp because I do not see how the reviewer's comments link to the present paper.

From the other reviewers, the authors have addressed the concerns raised by LGXe who has already give high scores for Soundness, Presentation and Contribution, so I believe the reviewer would likely have raised their score.

Reviewers 2x6h and cjJ7 already has positive ratings for the paper. cjJ7's single weakness of existing baselines has been addressed only very superficially. Most of 2x6h's weaknesses have been addressed.

Overall, I find it likely that the three included reviewers would have scored the paper positively and believe it meets the bar for acceptance.

---

### Decision · Program_Chairs · 2026-01-26

Accept (Poster)